# The Warburg Effect and lactate signaling augment Fgf-MAPK to promote sensory-neural development in the otic vesicle

Husniye Kantarci, Yunzi Gou, Bruce B Riley*

Biology Department, Texas A&M University, College Station, United States

**Abstract** Recent studies indicate that many developing tissues modify glycolysis to favor lactate synthesis (Agathocleous et al., 2012; Bulusu et al., 2017; Gu et al., 2016; Oginuma et al., 2017; Sá et al., 2017; Wang et al., 2014; Zheng et al., 2016), but how this promotes development is unclear. Using forward and reverse genetics in zebrafish, we show that disrupting the glycolytic gene *phosphoglycerate kinase-1* (*pgk1*) impairs Fgf-dependent development of hair cells and neurons in the otic vesicle and other neurons in the CNS/PNS. Fgf-MAPK signaling underperforms in *pgk1-/-* mutants even when Fgf is transiently overexpressed. Wild-type embryos treated with drugs that block synthesis or secretion of lactate mimic the *pgk1-/-* phenotype, whereas *pgk1-/-* mutants are rescued by treatment with exogenous lactate. Lactate treatment of wild-type embryos elevates expression of Etv5b/Erm even when Fgf signaling is blocked. However, lactate's ability to stimulate neurogenesis is reversed by blocking MAPK. Thus, lactate raises basal levels of MAPK and Etv5b (a critical effector of the Fgf pathway), rendering cells more responsive to dynamic changes in Fgf signaling required by many developing tissues.

*For correspondence:
briley@bio.tamu.edu

Competing interests: The authors declare that no competing interests exist.

## Introduction

Development of the paired sensory organs of the head relies on critical contributions from cranial placodes. Of the cranial placodes, the developmental complexity of the otic placode is especially remarkable for producing the entire inner ear, with its convoluted epithelial labyrinth and rich cell type diversity. The otic placode initially forms a fluid filled cyst, the otic vesicle, which subsequently undergoes extensive proliferation and morphogenesis to produce a series of interconnected chambers containing sensory epithelia (*Whitfield, 2015*). Sensory epithelia comprise a salt-and-pepper pattern of sensory hair cells and support cells. Hair cell specification is initiated by expression of the bHLH factor Atonal Homolog 1 (Atoh1) (*Bermingham et al., 1999*; *Chen et al., 2002*; *Millimaki et al., 2007*; *Raft et al., 2007*; *Woods et al., 2004*). Atoh1 subsequently activates expression of Notch ligands that mediate specification of support cells via lateral inhibition. Hair cells are innervated by neurons of the statoacoustic ganglion (SAG), progenitors of which also originate from the otic vesicle. Specification of SAG neuroblasts is initiated by localized expression of the bHLH factor Neurogenin1 (Ngn1) (*Andermann et al., 2002*; *Korzh et al., 1998*; *Ma et al., 1998*; *Raft et al., 2007*). A subset of SAG neuroblasts delaminate from the otic vesicle to form 'transit-amplifying' progenitors that slowly cycle as they migrate to a position between the otic vesicle and hindbrain before completing differentiation and extending processes to hair cells and central targets in the brain (*Alsina et al., 2004*; *Kantarci et al., 2016*; *Vemaraju et al., 2012*).

Development of both neurogenic and sensory domains of the inner ear requires dynamic regulation of Fgf signaling. Fgf-dependent induction of Ngn1 establishes the neurogenic domain during placodal stages, marking one of the earliest molecular asymmetries in the otic placode in chick and mouse embryos (*Abelló et al., 2010*; *Alsina et al., 2004*; *Magariños et al., 2010*; *Mansour et al., 1993*; *Pirvola et al., 2000*). Subsequently, as neurogenesis subsides, Fgf also specifies sensory

domains of Atoh1 expression (*Hayashi et al., 2008*; *Ono et al., 2014*; *Pirvola et al., 2002*). Moreover, in the mammalian cochlea different Fgf ligand-receptor combinations fine-tune the balance of hair cells and support cells (*Hayashi et al., 2007*; *Jacques et al., 2007*; *Mansour et al., 2009*; *Pirvola et al., 2000*; *Puligilla et al., 2007*; *Shim et al., 2005*). In zebrafish, sensory domains form precociously during the earliest stages of otic induction in response to Fgf from the hindbrain and subjacent mesoderm (*Gou et al., 2018*; *Millimaki et al., 2007*). Ongoing Fgf later specifies the neurogenic domain in an abutting domain of the otic vesicle (*Kantarci et al., 2015*; *Vemaraju et al., 2012*). As otic development proceeds, the overall level of Fgf signaling increases, reflected by increasing expression of numerous genes in the Fgf synexpression group, including transcriptional effectors Etv4 (Pea3) and Etv5b (Erm), and the feedback inhibitor Spry (Sprouty1, 2 and 4). Sensory epithelia gradually expand during development and express Fgf, contributing to the general rise in Fgf signaling. Similarly, mature SAG neurons express Fgf5. As more neurons accumulate, rising levels of Fgf eventually exceed an upper threshold to terminate further specification of neuroblasts and delay differentiation of transit-amplifying SAG progenitors (*Vemaraju et al., 2012*). Additionally, elevated Fgf inhibits further hair cell specification while maintaining support cells in a quiescent state (*Bermingham-McDonogh et al., 2001*; *Jiang et al., 2014*; *Ku et al., 2014*; *Maier and Whitfield, 2014*). Thus, dynamic changes in Fgf signaling regulate the onset, amount, and pace of neural and sensory development in the inner ear.

There is increasing evidence that dynamic regulation of glycolytic metabolism is critical for proper development of various tissues. For example, populations of proliferating stem cells or progenitors, including human embryonic stem cells, neural stem cells, hematopoietic stem cells, and posterior presomitic mesoderm, modify glycolysis by shunting pyruvate away from mitochondrial respiration in favor of lactate synthesis, despite an abundance of free oxygen (*Agathocleous et al., 2012*; *Bulusu et al., 2017*; *Gu et al., 2016*; *Oginuma et al., 2017*; *Sá et al., 2017*; *Wang et al., 2014*; *Zheng et al., 2016*). This is similar to 'aerobic glycolysis' (also known as the 'Warburg Effect') exhibited by metastatic tumors, thought to be an adaptation for accelerating ATP synthesis while simultaneously producing carbon chains needed for rapid biosynthesis (*Liberti and Locasale, 2016*). In addition, aerobic glycolysis and lactate synthesis appears to facilitate cell signaling required for normal development. For example, aerobic glycolysis promotes relevant cell signaling to stimulate differentiation of osteoblasts, myocardial cells, and fast twitch muscle (*Esen et al., 2013*; *Menendez-Montes et al., 2016*; *Ozbudak et al., 2010*; *Tixier et al., 2013*), as well as maintenance of cone cells in the retina (*Aït-Ali et al., 2015*).

In a screen to identify novel regulators of SAG development in zebrafish, we recovered two independent mutations that disrupt the glycolytic enzyme Phosphoglycerate Kinase 1 (Pgk1). Loss of Pgk1 causes a delay in upregulation of Fgf signaling in the otic vesicle, causing a deficiency in early neural and sensory development. We show that Pgk1 is co-expressed with Fgf ligands in the otic vesicle, as well as in the olfactory epithelium and various clusters of neurons in cranial ganglia and the neural tube. Moreover, Pgk1 acts non-autonomously to promote Fgf signaling at a distance by promoting synthesis and secretion of lactate. Lactate independently activates the MAPK pathway, leading to elevated expression of the effector Etv5b, priming the pathway to render cells more responsive to dynamic changes in Fgf levels.

## Results

### Initial characterization of *sagd1*

To identify novel genes required for SAG development, we conducted an ENU mutagenesis screen for mutations that specifically alter the number or distribution of post-mitotic *isl2b:Gfp+* SAG neurons. A recessive lethal mutation termed *sagd1* (*SAG deficient1*) was recovered based on a deficiency in the anterior/vestibular portion of ganglion, which are the first SAG neurons to form. Quantification of *isl2b:Gfp+* cells in serial sections of *sagd1-/-* mutants revealed a ~ 60% deficiency in anterior SAG neurons (which are essential for vestibular function) at 24 and 30 hpf (*Figure 1A,B*). The number of anterior neurons remained lower than normal through at least 48 hpf whereas accumulation of posterior SAG neurons (which includes auditory neurons) appeared normal (*Figure 1B*). Staining with anti-Isl1/2 antibody, which labels a more mature subset of SAG neurons, showed a similar trend: There was a 60% deficiency in anterior neurons from the earliest stages of SAG

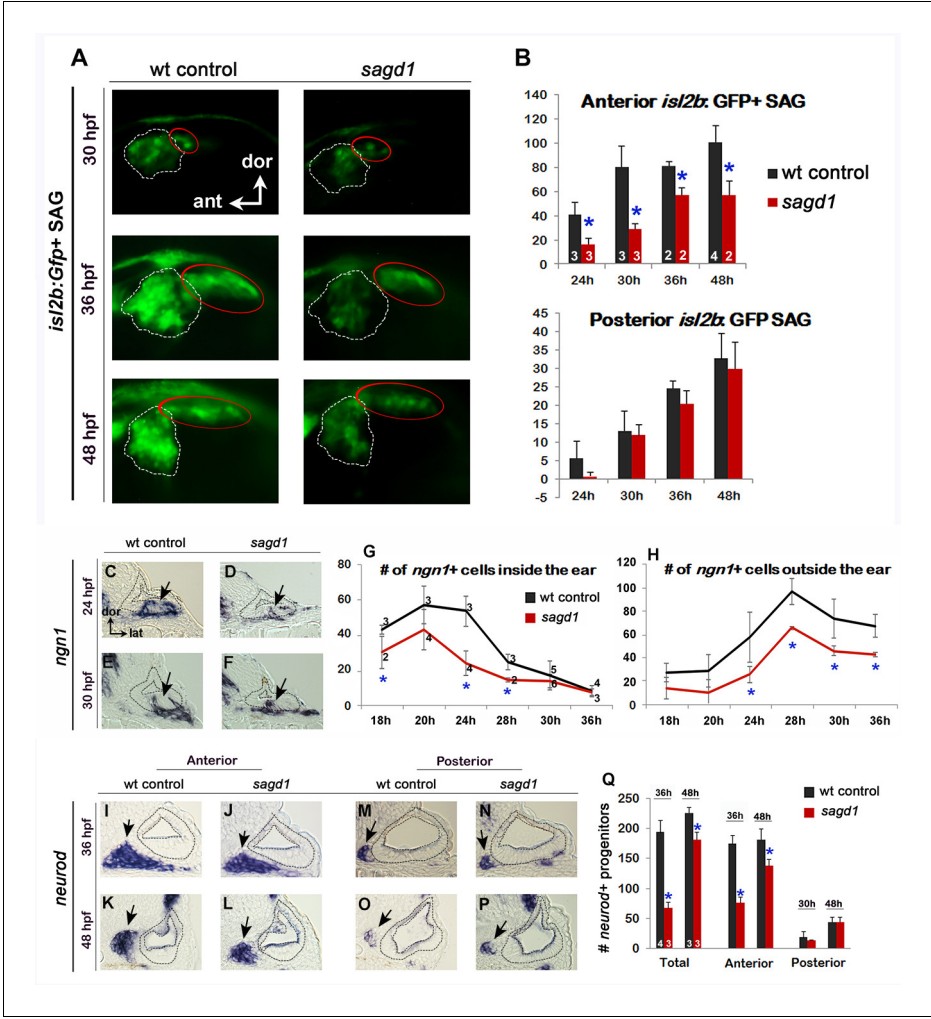

**Figure 1.** Initial analysis of *sagd1*. (**A**) Lateral views of *isl2b:Gfp*+ SAG neurons in live wild-type (wt) embryos and *sagd1* mutants at the indicated times. The anterior/vestibular portion of the SAG (outlined in white) is deficient in *sagd1* mutants, whereas the posterior SAG (outlined in red) appears normal. (**B**) Number (mean and s.d.) of *isl2b: Gfp*+ anterior neurons and posterior neurons in wild-type and *sagd1* embryos at the times indicated. Sample sizes are indicated. Asterisks, here and in subsequent figures, indicate significant differences (p<0.05) from wild-type controls. (**C–F**) Cross-sections through the anterior/vestibular portion of the otic vesicle (outlined) showing expression of *ngn1* in wild-type embryos and *sagd1* mutants at 24 and 30 hpf. (**G, H**) Mean and standard deviation of *ngn1*+ cells in the floor of the otic vesicle (**G**) and in recently delaminated SAG neuroblasts outside the otic vesicle (**H**), as counted from serial sections. (**I–P**) Cross-sections through the anterior/vestibular and posterior/ auditory regions of the otic vesicle showing expression of *neurod* in transit-amplifying SAG neuroblasts at 36 and 48 hpf. (**Q**) Mean and standard deviation of *neurod*+ SAG neuroblasts at 30 and 48 hpf counted from serial sections. Sample sizes are indicated (**B, G, Q**).

The online version of this article includes the following figure supplement(s) for figure 1:

**Figure supplement 1.** Quantitation of mature Isl1/2+ SAG neurons and TUNEL in *sagd1* mutants.

development whereas subsequent formation of posterior neurons was normal (*Figure 1—figure supplement 1A–I*). Of note, the early deficit of SAG neurons did not result from increased apoptosis (*Figure 1—figure supplement 1J*), and in fact *sagd1- / -* mutants produced fewer apoptotic cells than normal. *sagd1- / -* mutants show no overt defects in gross morphology, although *sagd1- / - mutants do show deficits in balance and motor coordination indicative of vestibular dysfunction. Mutants typically die by 10–12 dpf.

To further characterize the *sagd1- / -* phenotype, we examined earlier stages of SAG development. Specification of SAG neuroblasts is marked by expression of *ngn1* in the floor of the otic

vesicle, a process that normally begins at 16 hpf, peaks at 24 hpf, and then gradually declines and ceases by 42 hpf (*Vemaraju et al., 2012*). We observed that early stages of neuroblast specification were significantly impaired in *sagd1- / -* mutants, with only 30% of the normal number of *ngn1+* neuroblasts inside the otic vesicle at 24 hpf (*Figure 1C,D,G*). However, subsequent stages of neural specification gradually improved in *sagd1- / -* mutants, such that the number of *ngn1+* neuroblasts in the otic vesicle was normal by 30 hpf (*Figure 1E,F,G*). In the next stage of SAG development, a subset of SAG neuroblasts delaminates from the otic vesicle and enters a transit-amplifying phase. Recently delaminated neuroblasts continue to express *ngn1* for a short time before shifting to expression of *neurod*, which is then maintained until SAG progenitors mature into post-mitotic neurons. In *sagd1- / -* mutants, the number of *ngn1+* neuroblasts outside the otic vesicle was reduced by 50–60% at every stage examined (*Figure 1H*). Similarly, the number of *neurod+* transit-amplifying cells was strongly reduced in *sagd1- / -* mutants (*Figure 1I–Q*). This deficit was more pronounced at 30 hpf (~60% decrease) than at 48 hpf (~20% decrease). Moreover, deficit was restricted to anterior transit-amplifying cells that give rise to anterior neurons, which are the first to form, whereas accumulation of later-forming posterior progenitors was normal (*Figure 1Q*). In summary, the neural deficit observed in *sagd1- / -* mutants reflects a deficiency in specification of early neuroblasts that normally form the anterior SAG, which is essential for vestibular function. In addition, the continuing deficit in anterior transit-amplifying cells at later stages likely contributes to a long-term deficit in anterior SAG neurons.

We next examined other aspects of otic development in *sagd1- / -* mutants. Expression of prosensory marker *atoh1a* was reduced in both anterior (utricular) and posterior (saccular) maculae at 24 and 30 hpf (*Figure 2A–B'*). In agreement, *sagd1- / -* mutants produced roughly 25% fewer than normal hair cells in the utricle and saccule at 36 and 48 hpf (*Figure 2C–E*). Because development of SAG progenitors and sensory epithelia both rely on Fgf signaling, we also examined expression of various Fgf ligands and downstream Fgf-target genes. Expression of *fgf3* and *fgf8a* appeared

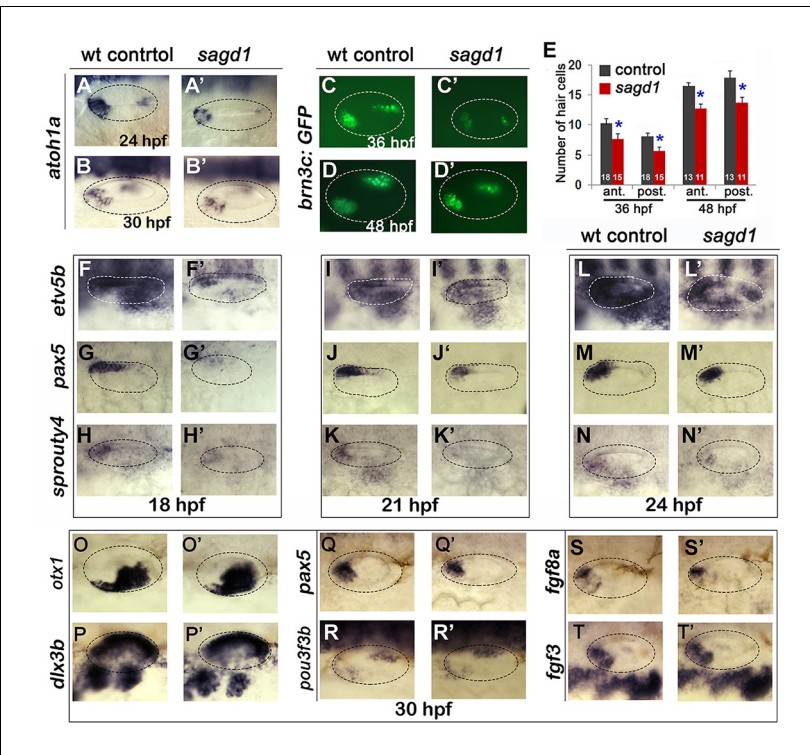

**Figure 2.** Sensory development and early Fgf signaling are impaired in *sagd1* mutants. (A–D", F–T') Dorsolateral views of whole mount specimens (anterior to the left) showing expression of the indicated genes in the otic vesicle (outlined) at the indicated times in wild-type embryos and *sagd1* mutants. (E) Mean and standard deviation of hair cells in the anterior/utricular and posterior/auditory maculae at 36 and 48 hpf in wild-type embryos (black) and *sagd1* mutants (red). Sample sizes are indicated.

relatively normal in *sagd1- / -* mutants from 18 to 30 hpf (*Figure 2S–T'*, and data not shown). However, expression of downstream Fgf-targets *etv5b*, *pax5* and *sprouty4* were severely deficient in the otic vesicle of *sagd1- / -* mutants at 18 hpf (*Figure 2F–H'*). Expression of these genes partially recovered by 21 hpf and was only mildly reduced by 24 hpf (*Figure 2I–N'*). Despite these changes, markers of axial identity were largely unaffected at 30 hpf, including the dorsal marker *dlx3b*, the ventrolateral marker *otx1*, the anteromedial marker *pax5* and posteromedial marker *pou3f3b* (*Figure 2O–R'*). These data suggest that Fgf signaling is impaired during early stages of otic vesicle development but slowly improves during later stages, which could explain the early deficits in sensory and neural specification. However, recovery of Fgf signaling is apparently incomplete or insufficient since production of new hair cells in *sagd1* mutants continues at a reduced rate through at least 48 hpf.

## Identification of the *sagd1* locus: A novel role for Pgk1

To identify the affected locus in *sagd1- / -* mutants, we used whole genome sequencing and homozygosity mapping approaches (*Obholzer et al., 2012*). We identified as our top candidate a novel transcript associated with the gene encoding the glycolytic enzyme Phosphoglycerate Kinase-1 (Pgk1) (*Figure 3—figure supplement 1*).

The zebrafish genome harbors only one *pgk1* gene, but the locus produces two distinct transcripts (*Figure 3A*). The primary transcript (*pgk1-FL*) encodes full-length Pgk1, which is highly conserved amongst vertebrates (nearly 90% identical between zebrafish and human). However, the second *pgk1* transcript, termed *pgk1-alt*, is unique to zebrafish and arises from an independent transcription start site in the first intron of *pgk1-FL*. The expression and sequence of *pgk1-alt* transcript were confirmed by conducting RT-PCR on mRNA harvested from wild-type embryos at 24 hpf (*Figure 3—figure supplement 2*). There are two short exons (termed exons 1a and 1b) at the start of *pgk1-alt* that encode a novel peptide with either 17 or 31 amino acids, depending on translation start site (see below). *pgk1-alt* then splices in-frame with exons 2–6, which are identical to *pgk1-FL*, followed by a splice into a novel exon (termed exon 6a) containing an in-frame stop codon (*Figure 3A*). Exon 6a then splices into exon 7, after which the sequence is identical to *pgk1-FL*. Thus *pgk1-alt* encodes a truncated protein that includes much of the N-terminal half of Pgk1 but presumably lacks glycolytic enzyme activity as the C-terminal peptide is required for ADP/ATP binding. In *sagd1- / -* mutants, *pgk1-alt* contains two nucleotide substitutions leading to loss of a splice acceptor site as well as the translation start codon in exon 1b (*Figure 3B*). Either of these SNPs could potentially disrupt expression of *pgk1-alt* protein. *pgk1-alt* transcript abundance is strongly reduced in *sagd1* mutants (*Figure 3—figure supplement 2*).

To determine which SNP in *sagd1* is most relevant, we tested morpholinos (MOs) designed to mimic either mutation. Injecting wild-type embryos with a splice-blocker to target the exon 1b splice acceptor site (sbMO, *Figure 3—figure supplement 3A*) impaired accumulation of *pgk1-alt* transcript (*Figure 3—figure supplement 3B*) and reduced accumulation of Isl1+ SAG neurons by nearly 40% (*Figure 3—figure supplement 3C*). To design translation blockers (tbMOs), we considered that exon1b contains two in-frame AUG codons that could potentially serve as translation start sites, although the first site (disrupted in *sagd1*) lacks a Kozak consensus. We therefore designed tbMO1 and tbMO2 to target each start site (*Figure 3—figure supplement 3A*). It has been established that MOs can effectively block translation by binding mRNA near the start site and up to 80 nucleotides upstream, whereas binding >10 nucleotides downstream is ineffective (gene-tools.com). Therefore, tbMO1 is expected block translation from either start site whereas tbMO2 is predicted to block only the second start site. Injecting either translation blocker alone, or in combination, reduced accumulation of Isl1+ SAG neurons by 20–30% (*Figure 3—figure supplement 3C*). The phenotype produced by tbMO2 suggests that the second ATG is more likely to serve as the translation start site, in which case altering the first AUG should not disrupt *pgk1-alt* function. Thus, the first SNP in *sagd1* that disrupts the exon 1b splice acceptor is the likely cause of the mutant phenotype. Interestingly, the translation blockers also impaired accumulation of *pgk1-alt* transcript (*Figure 3—figure supplement 3B*), suggesting that Pgk1-alt protein is required for normal transcript accumulation.

Although *sagd1* does not alter the sequence of *pgk1-FL*, we examined whether the *sagd1* mutation alters accumulation of *pgk1-FL* transcript. In wild-type embryos *pgk1-FL* transcript (but not *pgk1-alt*) is maternally deposited and is widely expressed at a low level during subsequent stages of development (*Figure 3—figure supplement 4A*). By 14 hpf *pgk1-FL* upregulates in developing

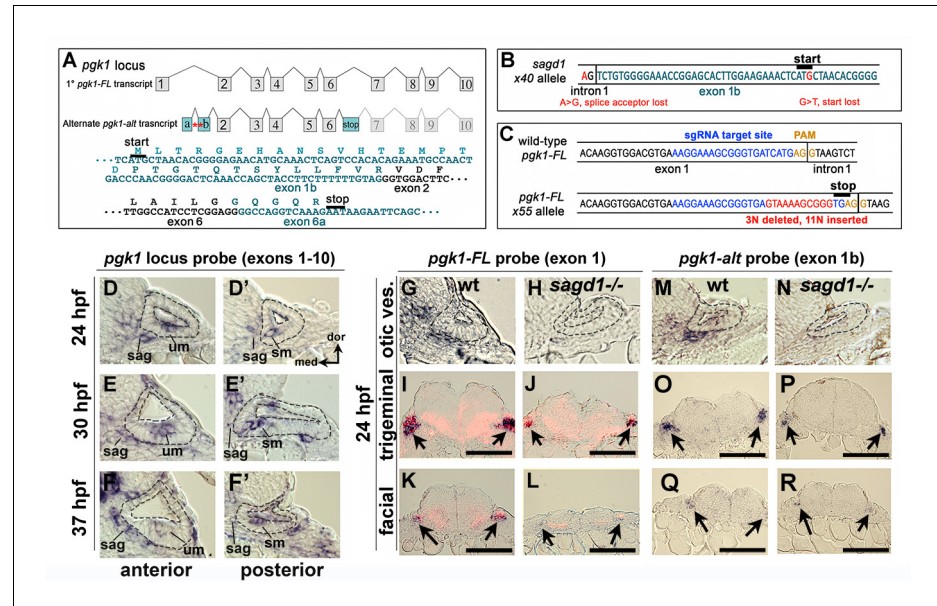

**Figure 3.** Two independent transcripts associated with the *pgk1* locus. (**A**) Exon-intron structure of the primary full-length *pgk1* transcript (*pgk1-FL*) and alternate transcript (*pgk1-alt*) arising from an independent transcription start site containing two novel exons (1a and 1b) that splice in-frame to exon 2, and a third novel exon (6a) containing a stop codon. The nucleotide and peptide sequences of exons 1b and 6a are shown. Relative positions of the lesions in *sagd1* affecting exon 1b are indicated (red asterisks). (**B**) Nucleotide sequence near the 5' end of exon 1b showing the SNPs detected in *sagd1* (red font). (**C**) Nucleotide sequence of *pgk1-FL* showing the sgRNA target site (blue font) and the altered sequence of the *x55* mutant allele (red font), which introduces a premature stop codon. (**D–F'**) Cross sections through the otic vesicle (outlined) showing staining with ribo-probe for the entire *pgk1* locus (covering both *pgk1-FL* and *pgk1-alt*) in wild-type embryos. Note elevated expression in the SAG, utricular macula (um), and saccular macula (sm). (**G, H, M, N**) Cross sections through the otic vesicle (outlined) in wild-type embryos and *sagd1* mutants stained with ribo-probe for exon 1 (*pgk1-FL* alone) (G, H) or exon 1b (*pgk1-alt* alone) (M, N). Accumulation of both transcripts is dramatically reduced in *sagd1* mutants. (**I–L, O–R**) Cross sections through the hindbrain (dorsal up) showing expression of *pgk1-FL* (black) plus *neurod* (red) (I–L) or *pgk1-alt* alone (O–R). Arrows indicate positions of the trigeminal and facial ganglia. Scale bar, 100 µm.

The online version of this article includes the following figure supplement(s) for figure 3:

**Figure supplement 1.** Mapping of SNPs linked to *sagd1*.
**Figure supplement 2.** RT-PCR identification of *pgk1-alt*.
**Figure supplement 3.** Morpholinos for *pgk1-alt* phenocopy *sagd1*.
**Figure supplement 4.** Expression of *pgk1-FL* during development.
**Figure supplement 5.** *pgk1-alt* misexpression increases *pgk1* transcript abundance.

---

somites (*Figure 3—figure supplement 4B*), and beginning around 18–20 hpf *pgk1-FL* transcript abundance becomes elevated in clusters of cells scattered throughout the central and peripheral nervous systems (*Figure 3—figure supplement 4C–E*). Examples of cells with elevated *pgk1-FL* expression include utricular and saccular hair cells, mature SAG neurons, trigeminal and epibranchial ganglia, the midbrain-hindbrain border, reticulospinal neurons in the hindbrain, and the olfactory epithelium (*Figure 3D–G,I,K* and *Figure 3—figure supplement 4D–G*). A similar pattern is observed for accumulation of *pgk1-alt* transcript (*Figure 3M,O,Q*). Local upregulation of *pgk1-FL* and *pgk1-alt* is severely attenuated in the nervous system in *sagd1- / -* mutants (*Figure 3D–R*), whereas expression remains elevated in somites (data not shown).

To test whether Pgk1-alt function is sufficient to upregulate *pgk1-FL*, we examined the effects of transient misexpression using a heat shock-inducible transgene, *hs:pgk1-alt*. When *hs:pgk1-alt* was activated at 20 hpf, accumulation of *pgk1-FL* was markedly elevated at 23–24 hpf (*Figure 3—figure supplement 5I–N*). This was in contrast to accumulation of transgenic *pgk1-alt* transcript, which peaked near the end of the heat shock period at 21 hpf but then decayed to background levels by 24 hpf (*Figure 3—figure supplement 5A–H*). We infer that perdurance of Pgk1-alt protein is

sufficient to upregulate *pgk1-FL*. Based on previous studies of yeast and mammalian Pgk1 (*Beckmann et al., 2015*; *Ho et al., 2010*; *Liao et al., 2016*; *Ruiz-Echevarria et al., 2001*; *Shetty et al., 2010*; *Shetty et al., 2004*), it is possible that Pgk1-alt acts to stabilize *pgk1-FL* transcript (See Discussion).

To directly test the requirement for full length Pgk1, we targeted the first exon of *pgk1-FL* using CRISPR-Cas9 and recovered an indel mutation leading to a frameshift followed by a premature stop (*Figure 3C*). This presumptive null mutation is predicted to eliminate all but the first 19 amino acids of Pgk1. Expression of *pgk1* is severely reduced in *pgk1- / -* mutants, presumably reflecting nonsense-mediated decay (*Figure 3—figure supplement 4H*). The phenotype of *pgk1- / -* mutants is nearly identical to *sagd1-/-*: Despite apparently normal early expression of *fgf3* and *fgf8a*, *pgk1- / -* mutants show impaired responses to early Fgf signaling, including reduced otic expression of *pax5* and *etv5b* at 18 hpf, reduced *ngn1* at 24 hpf, a deficiency in *neurod*+ transit-amplifying cells at 24 hpf, and a deficiency in mature *isl2b-gfp*+ SAG neurons at 30 hpf (*Figure 4A–H,Q*; *Figure 4—figure supplement 1*). The deficiency of Isl1+ SAG neurons persisted through at least 5 dpf (*Figure 4—figure supplement 1*). Unlike *sagd1* mutants, however, *pgk1- / -* mutants showed a deficiency in both anterior and posterior SAG neurons (reduced 31% and 18%, respectively, at 5 dpf, *Figure 4—figure supplement 1*). *pgk1- / -* mutants also showed a 20–25% decrease in the number of utricular and saccular hair cells at 36 hpf, 48 hpf and five dpf (*Figure 4T*). Other aspects of otic vesicle patterning appeared normal based on expression of regional markers *otx1*, *pou3f3b*, *hmx3a* and *dlx3b* at 24 hpf (*Figure 4—figure supplement 2*). Beyond the inner ear, development of a number of other Fgf-dependent neural cell types are also deficient in *pgk1- / -* mutants, including trigeminal, facial and glossopharyngeal ganglia, reticulospinal neurons in the hindbrain, and the olfactory epithelium (*Figure 4—figure supplement 3*). Although *pgk1- / -* mutants show overtly normal gross morphology, they eventually die around 10–12 days post-fertilization. When *pgk1+ / -* heterozygotes were crossed with *sagd1+ / -* heterozygotes, all intercross embryos developed normally (*Figure 4Q*), confirming that *pgk1-alt* and *pgk1-FL* encode distinct complementary functions. We also generated a heat-shock inducible transgene to overexpress *pgk1-FL* (*hs:pgk1*). Activation of *hs:pgk1* at 18 hpf rescued the SAG deficiency in *sagd1- / -* mutants (*Figure 4Q*). In contrast, activation of *hs:pgk1-alt* at 18 hpf was not able to rescue *pgk1- / -* mutants (*Figure 4—figure supplement 4A*), whereas activation of *hs:pgk1-alt* at 18 hpf did rescue morphants injected with *pgk1-alt*-sbMO (*Figure 4—figure supplement 4B*). Together, these data show that *pgk1-alt* is required upstream to locally upregulate *pgk1-FL*, which in turn is required for proper Fgf signaling and development of sensory hair cells, SAG neurons and various other Fgf-dependent neurons in the head.

## Pgk1 acts non-autonomously to promote fgf signaling

We note that sites of upregulation of *pgk1-FL* and *pgk1-alt* occurs within or near sites of expression of various Fgf ligands, including sensory epithelia, mature SAG neurons, and other tissues depicted in *Figure 3—figure supplement 4F–G* (*Reifers et al., 1998*; *Thisse et al., 2001*; *Millimaki et al., 2007*; *Feng and Xu, 2010*; *Terriente et al., 2012*; *Vemaraju et al., 2012*). Since expression of *fgf3* and *fgf8a* does not depend on *pgk1* (*Figure 2S-T*, *Figure 4I-J*, *Figure 4—figure supplement 1*), we asked whether upregulation of *pgk1* depends on Fgf. We found that local upregulation of *pgk1* occurred normally in embryos treated with the Fgf pathway-inhibitor SU5402 (*Figure 4—figure supplement 5*). Thus, *pgk1* and *fgf* genes are not required for each other's expression but could share a common upstream regulator.

To better understand how Pgk1 promotes Fgf signaling during otic development, we generated genetic mosaics by transplanting wild-type donor cells into *pgk1- / -* mutant host embryos. We reasoned that if Pgk1 acts cell-autonomously, as expected for a glycolytic enzyme, then isolated wild-type cells should be able to respond to local Fgf sources within a *pgk1- / -* mutant host. Surprisingly, the majority of the wild-type cells located near the utricular Fgf source in the floor of the otic vesicle in *pkg1- / -* hosts did not express the Fgf-target *etv5b* (*Figure 4J,L,N,P,R*). In control experiments, wild-type cells transplanted into wild-type hosts showed normal expression of *etv5b* (*Figure 4I,K,M, O,R*). Thus, Pgk1 is required non-autonomously to promote Fgf signaling at a distance.

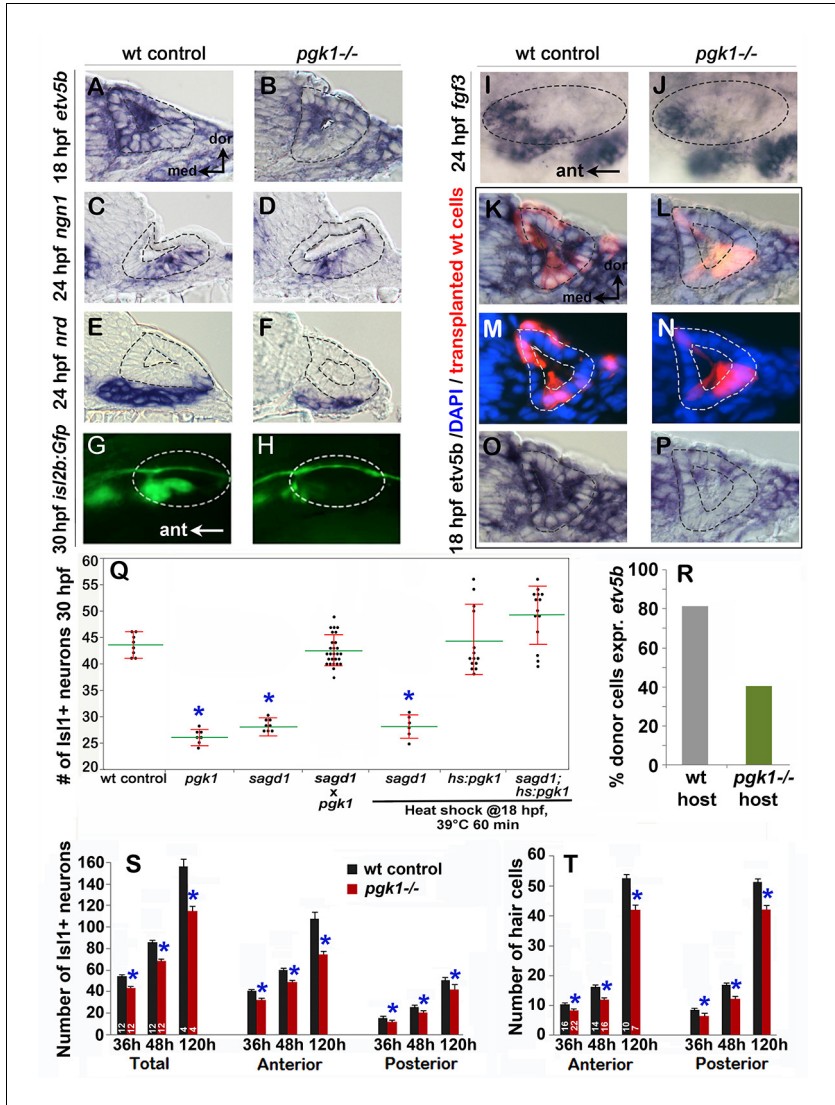

**Figure 4.** Initial characterization of *pgk1- / -* mutants and genetic mosaics. (A–F) Cross sections through the anterior/vestibular region of the otic vesicle (outlined) showing expression of the indicated genes in wild-type embryos and *pgk1- / -* mutants at the indicated times. (G–J) Lateral views showing *isl2b-Gfp* expression in live embryos at 30 hpf (G, H) and *fgf3* at 24 hpf (I, J). The otic vesicle is outlined. (K–P) Cross sections through the otic vesicle (outlined) showing positions of lineage labeled wild-type cells (red dye) transplanted into a wild-type host (K, M, O) or a *pgk1- / -* mutant host (L, N, P). Sections are co-stained with DAPI (M, N) and *etv5b* probe (O, P). Note the absence of *etv5b* expression in wild-type cells transplanted into the mutant host (P). (Q) Number of mature Isl1+ SAG neurons at 30 hpf in embryos with the indicated genotypes, except for *sagd1 x pgk1* intercross expected to contain roughly 25% each of *+/+*, *sagd1/+*, *pgk1/+* and *sagd1/pgk1* embryos. (R) Percent of wild-type donor cells located in the ventral half of the otic vesicle expressing *etv5b* in wild-type or *pgk1- / -* hosts. A total of 151 wild-type donor cells were counted in eight otic vesicles of wild-type host embryos, and 247 wild-type donor cells were counted in eight otic vesicles of *pgk1- / -* host embryos. (S) Number of Isl1+ SAG neurons (total, anterior and posterior) in wild-type and *pgk1- / -* mutant embryos at 36 hpf (n = 12), 48 hpf (n = 12) and 120 hpf (n = 4). (T) Number of phalloidin-stained hair cells (anterior/utricular and posterior/saccular) in wild-type and *pgk1- / -* mutant embryos at 36 hpf (n = 16), 48 hpf (n = 16) and 120 hpf (n = 10). Sample sizes are indicated (S, T). The online version of this article includes the following figure supplement(s) for figure 4:

**Figure supplement 1.** pgk1– / – mutants show normal onset of *fgf3* and *fgf8a*, but not *pax5*, in the otic vesicle.
**Figure supplement 2.** pgk1– / – mutants show normal expression of regional markers of the otic vesicle.
**Figure supplement 3.** Deficiencies in Fgf-dependent cell types in *pgk1- / -* mutants.
**Figure supplement 4.** Misexpression of *pgk1-alt* does not rescue *pgk1- / -* mutants but does rescue *pgk1-alt* morphants.

*Figure 4 continued on next page*

*Figure 4 continued*

**Figure supplement 5.** Fgf signaling is not required for normal *pgk1-FL* expression.

**Figure supplement 6.** Knockdown of *plasminogen* does not alter *pgk1-/-* or *sagd1-/-* phenotypes.

## Pgk1 does not act extracellularly through plasmin turnover

We considered two distinct mechanisms for non-autonomous functions of Pgk1 that have been documented in metastatic tumors. First, tumor cells secrete Pgk1 to promote early stages of metastasis through modulation of extracellular matrix (ECM) and cell signaling (*Chirico, 2011*; *Jung et al., 2009*; *Wang et al., 2007*; *Wang et al., 2010*). The only specific molecular function identified for secreted Pgk1 is to serve as a disulfide reductase leading to proteolytic cleavage of Plasmin (*Lay et al., 2000*), a serine protease that cleaves Fgf as well as ECM proteins required for Fgf signaling (*Botta et al., 2012*; *George et al., 2001*; *Meddahi et al., 1995*; *Schmidt et al., 2005*). This raised the possibility that loss of Pgk1 could elevate Plasmin activity and thereby impair Fgf signaling. To test this, we injected translation-blocking morpholino (*plg*-tbMO) to block synthesis of Plasminogen (Plg), the zymogen precursor of Plasmin. Injection of plg-tbMO did not rescue or ameliorate the neural deficiency in *sagd1-/-* or *pgk1-/-* mutants (*Figure 4—figure supplement 6A*). We also tested a splice-blocker (*plg*-sbMO) to validate MO efficacy. Injection of *plg*-sbMO effectively attenuated *plg* transcript (*Figure 4—figure supplement 6C*) but did not rescue *pgk1-/-* mutants (*Figure 4—figure supplement 6B*). Therefore the effects of *pgk1-/-* and *sagd1-/-* do not depend on accumulation of Plasmin.

## Aerobic glycolysis is required for otic development

The second Pgk1-dependent mechanism employed by metastatic tumors is to upregulate and redirect glycolysis to promote lactate synthesis despite abundant oxygen. This process is often called 'aerobic glycolysis', or the 'Warburg Effect', and is thought to accelerate synthesis of ATP and provide 3-carbon polymers used for rapid biosynthesis (*Liberti and Locasale, 2016*). In addition, lactate secretion facilitates cell signaling in cancer cells (*San-Millán and Brooks, 2017*) as well as during certain normal physiological contexts (*Lee et al., 2015*; *Peng et al., 2016*; *Yang et al., 2014*; *Zuo et al., 2015*; Reviewed by *Philp et al., 2005*). We therefore tested whether blocking glycolysis and/or lactate synthesis in wild-type embryos could mimic the phenotype of *pgk1-/-* and *sagd1-/-* mutants. Treating wild-type embryos with 2-deoxy-glucose (2DG) (*Nirenberg and Hogg, 1958*) or 3PO (*Clem et al., 2008*) to block early steps in glycolysis (*Figure 5A*) reduced the number of mature SAG neurons to a level similar to that of *pgk1-/-* or *sagd1-/-* mutants (*Figure 5B*). Similar results were obtained when these inhibitors were added at 0 hpf or 14 hpf.

We next focused on later steps in the pathway dealing with handling of pyruvate vs. lactate. In cells exhibiting the Warburg Effect, transport of pyruvate into mitochondria is typically blocked by upregulation of Pyruvate Dehydrogenase Kinase (PDK) (*Cairns et al., 2011*), thereby favoring reduction of pyruvate to lactate (see Warburg Shunt, *Figure 5A*). Dichloroacetate (DCA) blocks PDK activity (*Kato et al., 2007*), allowing more pyruvate to be transported into mitochondria and thereby forestalling lactate synthesis. Treating wild-type embryos with DCA at 14 hpf mimicked the deficiency of SAG neurons seen in *pgk1-/-* and *sagd1-/-* mutants (*Figure 5B*). To directly block synthesis of lactate from pyruvate, wild-type embryos were treated with Galloflavin, an inhibitor of Lactate Dehydrogenase (*Farabegoli et al., 2012*; *Manerba et al., 2012*; *Figure 5A*). Wild-type embryos treated with Galloflavin at 14 hpf also displayed a deficiency of SAG neurons similar to *pgk1-/-* and *sagd1-/-* mutants (*Figure 5C*). Finally, to block transport of lactate across the cell membrane, we treated wild-type embryos with either CHC or UK5099, which block activity of monocarboxylate transporters MTC1, 2 and 4 (*Halestrap, 1975*; *Halestrap and Denton, 1974*; *Figure 5A*). Application of either CHC or UK5099 at 14 hpf also mimicked the SAG deficiency in *pgk1-/-* and *sagd1-/-* mutants (*Figure 5C*, and data not shown). Treatment of wild-type embryos with 2DG, Galloflavin or CHC also reduced hair cell production to a level comparable to *sagd1-/-* or *pgk1-/-* mutants (*Figure 5D*). To further explore the role of lactate, we added exogenous lactate to embryos beginning at 14 hpf and examined accumulation of SAG neurons at 30 hpf. As shown in *Figure 5C*, lactate treatment elevated SAG accumulation above normal in wild-type embryos and fully rescued *pgk1-/-* mutants. Lactate treatment also rescued hair cell production in wild-type embryos treated

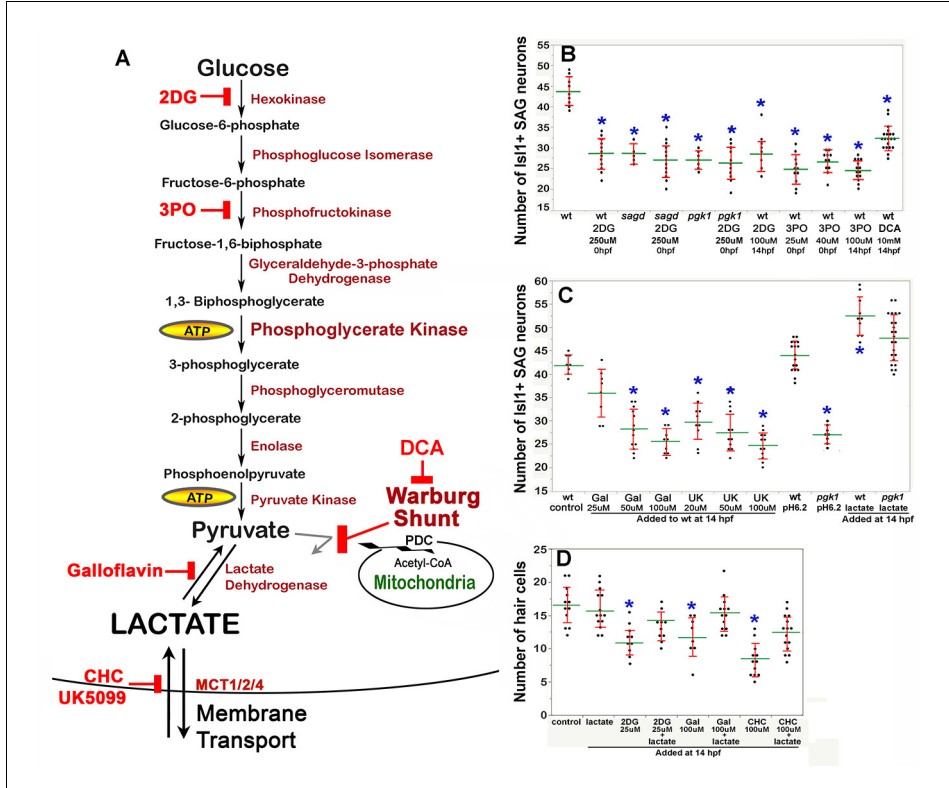

**Figure 5.** Drugs blocking aerobic glycolysis mimic *pgk1- / -* mutants. (**A**) Diagram of the glycolytic pathway and changes associated with the Warburg Effect in which pyruvate is shunted from mitochondria in favor of synthesis and secretion of lactate. The indicated inhibitors were used to block discrete steps in the pathway. In the Warburg Shunt, mitochondrial Pyruvate Dehydrogenase Complex (PDC) is inhibited by elevated Pyruvate Dehydrogenase Kinase (PDK), the activity of which is blocked by DCA. (**B, C**) Scatter plots showing the mean (green) and standard deviation (red) of the number of Isl1/2+ SAG neurons at 30 hpf in embryos with the indicated genotypes and/or drug treatments. Lactate-treatments and some controls were buffered with 10 mM MES at pH 6.2. (**D**) Scatter plats showing the mean and standard deviation of the total number of hair cells at 36 hpf in embryos treated with the indicated drugs. Asterisks show significant differences (p ≤. 05) from wild-type control embryos.

with 2DG, Galloflavin, or CHC (*Figure 5D*). Thus, pharmacological agents that block glycolysis, lactate synthesis or lactate secretion are sufficient to mimic the *pgk1- / -* and *sagd1- / -* phenotype, and exogenous lactate is sufficient to reverse these effects.

We next examined whether the above inhibitors and/or exogenous lactate affect Fgf signaling and SAG specification in the nascent otic vesicle. Treatment of wild-type embryos with Galloflavin or CHC reduced the number of cells expressing *etv5b* and *ngn1* at 18 hpf and 24 hpf, respectively (*Figure 6Ag*, Ah, Bg, Bh, E), mimicking the effects of *pgk1-/-* (*Figure 6Ab*, Bb, C, D). Treatment with exogenous lactate reversed the effects of Galloflavin and CHC (*Figure 6Ai, Aj, Bj*) and rescued *pgk1- / -* mutants (*Figure 6Ac*, Bc, C, D). Moreover, treating wild-type embryos with lactate increased the number of cells expressing *etv5*b and *ngn1* (*Figure 6Ae*, Be, C, D).

We also tested whether a deficiency of ATP contributes to the *pgk1- / -* phenotype. Adding exogenous ATP to *pgk1- / -* mutants at 14 hpf did not ameliorate the deficiency in expression of *etv5b*, *ngn1*, or *pax5* (*Figure 6Ad*, Bd, C, D; and *Figure 6—figure supplement 1*). Likewise, adding ATP to wild-type embryos at 14 hpf did not significantly affect expression of these genes (*Figure 6Af*, Bf, C, D; and *Figure 6—figure supplement 1*). Finally, adding 25 uM sodium azide to wild-type embryos at 14 hpf to block mitochondrial respiration did not significantly impair expression of *ngn1* at 24 hpf (*Figure 6Bk*, E). Thus, lactate is critical for early Fgf signaling and SAG specification, but exogenous ATP does not affect these functions. Moreover, the deficiencies in otic development observed in *pgk1- / -* mutants can be attributed to disruption of lactate synthesis and secretion.

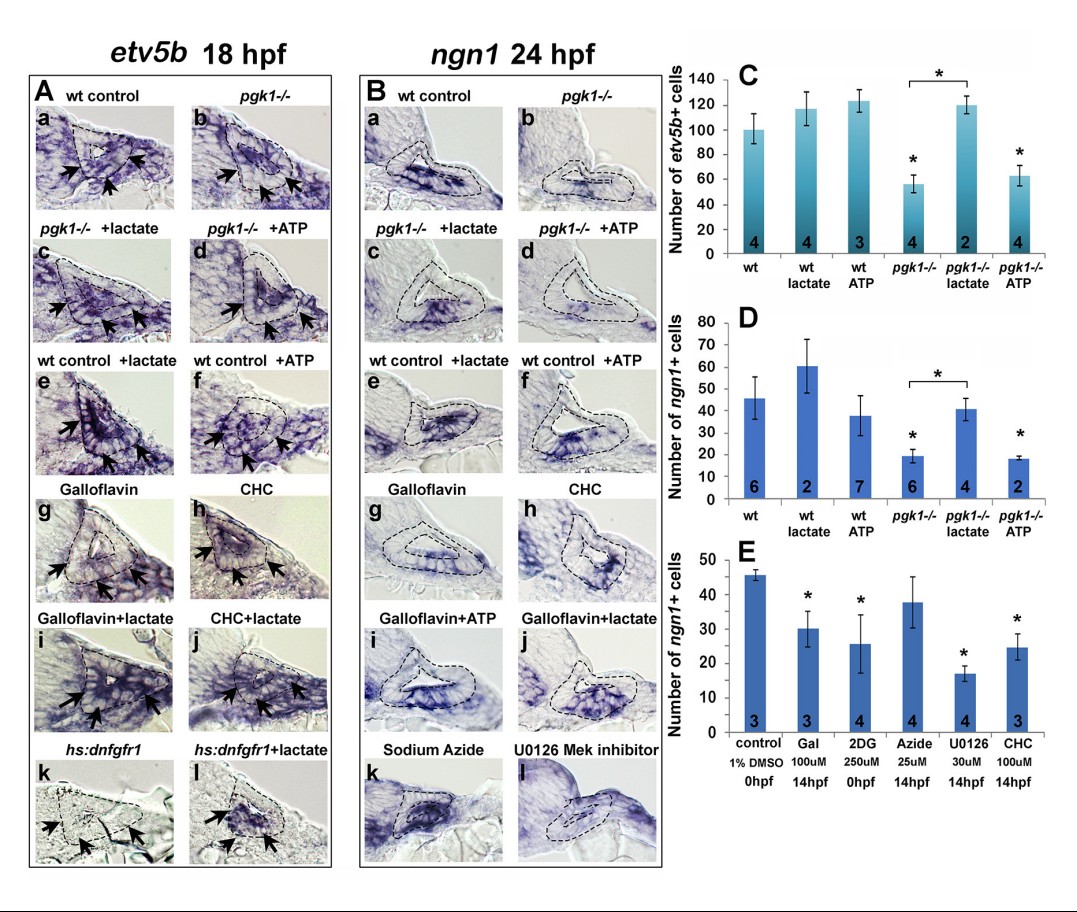

**Figure 6.** Exogenous lactate reverses the effects of disrupting aerobic glycolysis. (A, B) Cross sections through the anterior/vestibular portion of the otic vesicle (outlined) showing expression of *etv5b* at 18 hpf (A) and *ngn1* at 24 hpf (B) in embryos with the indicated genotype or drug treatment. Arrows in (A) highlight expression in the ventromedial otic epithelium. (C) Mean and standard deviation of the number of *etv5b+* cells in the ventral half of the otic vesicle counted from serial sections of embryos with the indicated genotype or drug treatment. (D, E) Mean and standard deviation of the number of *ngn1+* cells in the floor of the otic vesicle counted from serial sections of embryos with the indicated genotype or drug treatment. Asterisks indicate significant differences (p<0.05) from wild-type controls, or between groups indicated by brackets.

The online version of this article includes the following figure supplement(s) for figure 6:

**Figure supplement 1.** Exogenous ATP does not rescue *pgk1- / -* mutants.

**Figure supplement 2.** Effects of altering MAPK/Erk, lactate and Fgf on SAG development.

## Lactate and fgf act in parallel to regulate early otic development

Fgf regulates gene expression primarily through the PI3K and MAPK/Erk signal transduction pathways, but it is unknown which is most critical for otic development. Treatment of wild-type embryos with U0126, an inhibitor of the MAPK activator Mek (*Hawkins et al., 2008*), reduced the number of *ngn1+* cells at 24 hpf (*Figure 6Bl*, E) and mature Isl1+ SAG neurons at 30 hpf (*Figure 6—figure supplement 2A*) to a degree similar to or greater than *pgk1- / -* mutants. In contrast, treatment of wild-type embryos with the PI3K inhibitor LY294002 (*Montero et al., 2003*) had negligible effects on these genes (*Figure 6—figure supplement 2A*, and data not shown), showing that most of the effects of Fgf are mediated by MAPK/Erk.

Recent findings show that lactate can activate the MAPK/Erk pathway through Ras-independent activation of Raf kinase (*Lee et al., 2015*), see Discussion). Consistent with this mechanism, the ability of lactate to stimulate production of SAG neurons was completely blocked by co-treatment with U0126 (*Figure 6—figure supplement 2B*). We next investigated whether lactate treatment can bypass the requirement for Fgf signaling. Transient misexpression of dominant negative Fgf receptor

from a heat shock-inducible transgene, *hs:dnfgfr1*, fully suppressed *etv5b* expression within two hours of heat shock (*Figure 6Ak*; *Figure 7C*) and reduced accumulation of SAG neurons by 25% at 30 hpf (*Figure 6—figure supplement 2C*). Importantly, treatment with exogenous lactate partially restored *etv5b* expression when Fgf signaling was blocked (*Figure 6Al*; *Figure 7D*), although this was not sufficient to restore development of SAG neurons (*Figure 6—figure supplement 2C*). Because developmental defects in *pgk1- / -* and *sagd1- / -* mutants appear to stem from reduced Fgf signaling, we tested whether elevating Fgf can rescue *pgk1- / -* mutants. Activation of *hs:fgf8* at a moderate level (37°C for 30 min) beginning at 16 hpf increased the level of *pax5* expression in the otic vesicle at 18 hpf and expanded the spatial domain (*Figure 7E,G*). Activation of *hs:fgf8* in *pgk1- / -* mutants increased expression of *pax5* relative to *pgk1- / -* alone (*Figure 7F,H*), but not to the same degree seen in non-mutant embryos. Thus, elevating either Fgf or lactate alone can partially compensate for loss of the other, but full activation of early otic genes requires both Fgf and lactate to fully activate the MAPK/Erk pathway.

## Discussion

### Enhanced glycolysis and lactate secretion promote fgf signaling

Metabolic pathways such as glycolysis have traditionally been viewed as 'house keeping' functions but are increasingly recognized for their specific roles in development. We discovered that mutations in *pgk1* cause specific defects in formation of hair cells and SAG neurons of the inner ear, as well as the olfactory epithelium and various cranial ganglia and reticulospinal neurons of the hindbrain, all of which normally show elevated expression of *pgk1* (*Figure 3*) and require Fgf signaling (*Lassiter et al., 2009*; *Maulding et al., 2014*; *Millimaki et al., 2007*; *Nechiporuk et al., 2005*; *Terriente et al., 2012*; *Vemaraju et al., 2012*). Treatment of wild-type embryos with specific inhibitors of glycolysis or lactate synthesis or secretion mimicked the effects of *pgk1-/-*, indicating that the affected cell types require 'aerobic glycolysis', similar to the Warburg Effect exhibited by metastatic tumors. Developmental deficiencies seen in *pgk1- / -* mutants reflect weakened response to Fgf but are fully rescued by treatment with exogenous lactate. Lactate has been shown to bind and stabilize the cytosolic protein Ndrg3, which in turn activates Raf (*Lee et al., 2015*) thereby converging with Fgf signaling to activate MAPK/Erk. In support of this mechanism, the stimulatory effects of lactate are blocked by the Mek-inhibitor U0126. The MAPK/Erk pathway activates transcription of *etv4* and *etv5a/b* genes (*Raible and Brand, 2001*; *Roehl and Nüsslein-Volhard, 2001*). Etv4/5 proteins are also phosphorylated and activated by MAPK/Erk and mediate many of the downstream effects of

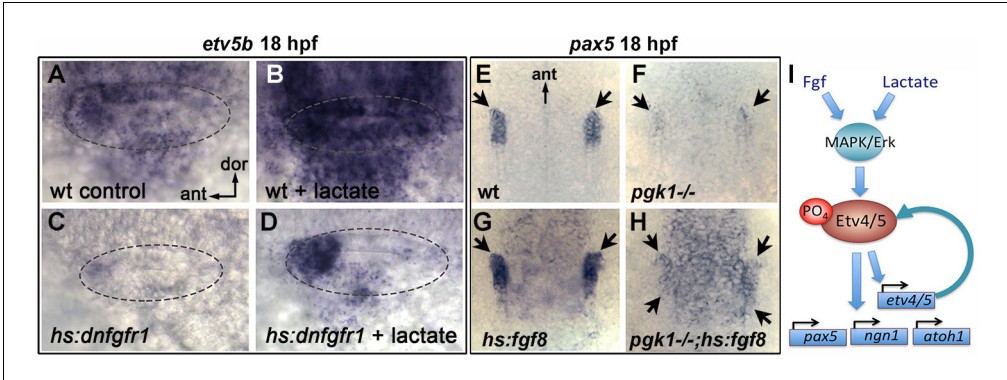

**Figure 7.** Fgf and lactate are required together for full activation of early otic genes. (**A–D**) Dorsolateral view of the otic vesicle (outlined) showing expression of *etv5b* at 18 hpf in wild-type embryos or *hs:dnfgfr1/+* transgenic embryos with or without exogenous 6.7 mM lactate added at 14 hpf, as indicated. Embryos were heat shocked at 39°C for 30 min beginning at 16 hpf. (**E–H**) Dorsal view (anterior to the top) of the hindbrain and otic region showing expression of the otic domain of *pax5* (black arrows) in embryos with the indicated genotypes. Embryos were heat shocked at 37°C for 30 min beginning at 16 hpf. (**I**) A model for how lactate and Fgf signaling converge on MAPK/Erk to increase the pool of phosphorylated activated Etv4/5, enabling cells to respond more quickly and robustly to dynamic changes in Fgf.

Fgf (*Brown et al., 1998*; *O'Hagan et al., 1996*; *Znosko et al., 2010*). This dual regulation of Etv4/5 implies a feedback amplification step and offers an explanation for why lactate is required to increase the efficiency of early stages of Fgf signaling (*Figure 7I*). Specifically, initial activation of MAPK/Erk by lactate would expand the pool of Etv4/5, priming cells to respond more quickly to dynamic changes in Fgf signaling. Without *pgk1*, the initial response to Fgf is sluggish and causes lasting deficits in SAG neurons and slows accumulation of hair cells. Later, as Fgf-mediated feedback amplification continues, Fgf signaling gradually improves on its own and is sufficient to support normal development of posterior SAG neurons in *sagd1* mutants, although accumulation of SAG neurons and hair cell continues at a slower pace in *pgk1- / -* mutants.

It is interesting that sites of *pgk1* upregulation correlate with sites of Fgf ligand expression. Fgf expression does not require *pgk1* (*Figure 2S–T'*), and *pgk1* upregulation does not require Fgf (*Figure 4—figure supplement 5*), suggesting co-regulation by shared upstream factor(s). Analysis of genetic mosaics showed that isolated wild-type cells transplanted into *pgk1- / -* host embryos failed to activate *etv5b* despite close proximity to sensory epithelia, a known Fgf-source. Clearly wild type cells retain the ability to perform glycolysis, but since upregulation of *pgk1* in the otic vesicle is limited to sensory epithelia and mature SAG neurons (*Figure 3A–G*), other otic cells may be unable to generate sufficient lactate to augment Fgf signaling. This raises the interesting possibility that Fgf and lactate must be co-secreted from signaling sources to support full signaling potential. Indeed, this model explains why SAG neuroblasts in the otic vesicle, which do not detectably express *pgk1* or *fgf* genes, require both to be expressed in nearby tissues for proper specification. Note, we cannot exclude the possibility that lactate acts cell-autonomously to promote secretion of Fgf ligand, but such a mechanism has never been reported and does not explain why blocking lactate transport mimics the effects of *pgk1- / -* and *sagd1* mutations.

## Regulation of Pgk1 by Pgk1-alt

The *sagd1* mutation does not directly affect full length Pgk1, but instead disrupts a novel transcript arising from an independent downstream transcription start site that appears to reflect an ancient transposon insertion. The first 17 amino acids of Pgk1-alt are novel but downstream the sequence is identical to exons 2–6 of Pgk1-FL. Production of Pgk1-alt is necessary and sufficient for upregulation of *pgk1-FL*, possibly involving transcript stabilization. Exon 6 of zebrafish Pgk1-FL and Pgk1-alt corresponds closely to the STE (stabilizing element) of yeast Pgk1 (*Ruiz-Echevarria et al., 2001*). Premature stops within the first half of yeast *Pgk1* lead to nonsense-mediated decay (NMD) of the transcript, but later stops (occurring downstream of the STE) do not trigger NMD (*Hagan et al., 1995*). The STE can also stabilize recombinant transcripts containing foreign sequences that include a 3'UTR that normally destabilizes the transcript independent of NMD. In each case the STE must be translated in order to function, raising the possibility that the corresponding peptide is required for stability (*Ruiz-Echevarria et al., 2001*). Several recent studies have identified full length Pgk1 in yeast and human as an unconventional mRNA-binding protein (*Beckmann et al., 2015*; *Liao et al., 2016*), and human Pgk1 can bind specific mRNAs to affect stability, though often by destabilization (*Ho et al., 2010*; *Shetty et al., 2004*). Whether Pgk1 increases or decreases mRNA stability could reflect interactions with specific sequences or secondary structures in the target. In any case, it seems reasonable that the STE of Pgk1-alt has been coopted from an established 'moonlighting' function of Pgk1 to locally upregulate *pgk1-FL* by increasing transcript stability, thereby facilitating the switch to Warburg metabolism.

Despite the requirement for *pgk1-alt* in the developing nervous system, it is not required in mesodermal tissues. For example, upregulation of *pgk1* in developing somites is not affected in *sagd1- / -* mutants. Curiously, *pgk1* is not appreciably upregulated in presomitic mesoderm, despite the fact that somitogenesis involves a gradient of aerobic glycolysis and lactate synthesis that is highest in the tail and declines anteriorly (*Bulusu et al., 2017*; *Oginuma et al., 2017*; *Ozbudak et al., 2010*). However, establishment of Warburg-like metabolism does not necessarily require elevated *pgk1*. Conversely, elevated *pgk1* does not necessarily indicate a Warburg-like state. Indeed, upregulation of *pgk1* in somites likely provides high levels of pyruvate to favor robust mitochondrial ATP synthesis over lactate synthesis (*Ozbudak et al., 2010*). Although we detected no obvious defects in somite formation in *pgk1- / -* or *sadg1- / -* mutants, it is possible that later aspects of somite differentiation, muscle physiology, etc. are affected.

Although the *sagd1* mutation was recovered in a forward ENU mutagenesis screen, we note that the SNPs detected in the *pgk1-alt* sequence were previously reported in the genome sequence database. It is therefore likely that familial breeding during our screen allowed generation and detection of homozygous carriers of this preexisting allele. Indeed, we have subsequently detected these SNPs in our wild-type stock, likely accounting for variation in expression levels in early otic genes sometimes detected in specific families. As a cautionary note, the sequence for *pgk1-alt* was initially represented in older versions of the annotated zebrafish genome but has since been removed, presumably deemed a technical artifact. However, RT-PCR amplification of sequences expressed at 24 hpf confirmed the presence of *pgk1-alt*. Had we used more recent versions of the genome as our reference dataset, the SNPs in *sagd1* would have been overlooked. Regardless of its origins, identification of the *sagd1* mutant allele underscores the continuing utility of forward screens: Had we not recovered *sagd1* from our screen, it is highly unlikely that we would have identified *pgk1* as a developmental regulatory gene.

## Materials and methods

### Fish strains and developmental conditions

Wild-type embryos were derived from the AB line (Eugene,OR). For most experiments embryos were maintained at 28.5℃, except where noted. Embryos were staged according to standard protocols (*Kimmel et al., 1995*). PTU (1-phenyl 2-thiourea), 0.3 mg/ml (Sigma P7629) was included in the fish water to inhibit pigment formation. The following transgenic lines were used in this study: Tg (*hsp70:fgf8a*)[x17] (*Millimaki et al., 2010*), Tg(*Brn3c:GAP43-GFP*)[s356t] (*Xiao et al., 2005*), Tg (−17.6*isl2b:GFP*)[zc7] (*Pittman et al., 2008*), Tg(*hsp70I:dnfgfr1-EGFP*)[pd1] (*Lee, 2005*), and new lines generated for this study Tg(*hsp70I:pgk1-FL*)[x65] and Tg(*hsp70I:pgk1-alt*)[x66]. These transgenic lines are herein referred to as *hs:fgf8*, *brn3c:GFP*, *Isl2b:GFP*, *hs:pgk1-FL*, *hs:pgk1-alt*, and *hs:dnfgfr1* respectively. Mutant lines *sagd1*[x40] and *pgk1*[x55] were used for loss of function analysis. Homozygous *sagd1- / -* and *pgk1- / -* mutant embryos were identified by a characteristic decrease in the expression of *Isl2b-Gfp* observed at stages after 24 hpf. At earlier stages, homozygous *pgk1- / -* mutant embryos were identified by indel-PCR genotyping using the following primers (5′−3′) in a single reaction: *pgk1*-fwd-1, AGGTCATTCTCATTCGGGAAC; *pgk1*-indel-1, <u>AAGGAAAGCGGGTGATCA TG</u>; *pgk1*-rev-1, ACGTTAAAGGGCATACGACG. *pgk1*-fwd-1 produces an amplicon of ~250 bp from both wild-type and mutant DNA, whereas *pgk1*-indel-1 produces an amplicon of 127 bp from wild-type DNA only. Adult *pgk1+ / -* heterozygotes were identified using the following primers in a single reaction: *pgk1*-fwd-2, <u>GCAAGTACATCCAATTGCCG</u>; *pgk1*-indel-2, CGGGTGAGTAAAAGCGGG; *pgk1*-rev-2, <u>ACGCGTGACAGATACACTTCC</u>. *pgk1*-fwd-2 produces an amplicon of ~451 bp from both wild type and mutant DNA, whereas *pgk1*-indel-2 produces an amplicon of 236 bp from mutant DNA only. *hs:pgk1*-FL and *hs:pgk1*-alt carriers were identified by in situ hybridization using probes for *pgk1*-FL and *pgk1*-alt transcripts following heat-shock at 39℃ for 1 hr, or by PCR genotyping using DNA extracted from tails of single embryos, and the following primers (5′−3′): hsp-Fwd, GACGAGGTGTTTATTCGCTCT; reverse primer, ACTGCAGCCTTGATTCTCTG, yielding an amplicon of ~700 bp.

### Gene misexpression and morpholino injections

Heat-shock regimens were carried in a water bath at indicated temperatures and durations. Embryos were kept in a 33℃ incubator after the heat-shock. Knockdown of *pgk1-alt* was performed by injecting wild-type embryos at the one-cell stage with ~5 ng of translation-blocking morpholino oligomer tbMO1 (5′-AGCATGAGTTTCTTCCAAGTGCACC-3′) or tbMO2 (5′-GTCAGTTGGCATTTCTGTG TGGACT-3′), or 10 ng of splice-blocker sbMO (5′-TCCCCACAGACTAAAGACAATCAGT-3′). Knockdown of *plasminogen* was performed by injecting one-cell embryos with ~5 ng of translation-blocker *plg*-tbMO (5′-AACTGCTTTGTGTACCTCCATGTCG-3′) or splice-blocker *plg*-sbMO (5′-AGCATCC TTACCTGTAAAAAGAAAG-3′). Although none of these knockdown conditions, by themselves, caused visible morphological changes or cell death, as a precaution all morpholinos were co-injected with ~5 ng *p53*-MO to suppress non-specific cell death.

## RNA extraction and RT-PCR

For *plg* RT-PCR, RNA was extracted from groups of 50 embryos at 30 hpf using Trizol (Life Technologies) and chloroform (MACRON fine chemicals). For *pgk1*-FL and *pgk1*-alt RT-PCR, RNA was extracted from groups of 50 wild type embryos at 2–4 cell stage or 24 hpf using Trizol and chloroform or a total RNA miniprep kit (NEB), including an on-column DNase I treatment. Trizol and chloroform extracted samples were treated with RQ1 DNase (Promega), and re-purified using a RNA cleanup kit (zymo). Reverse transcriptions were performed in 20 μl reactions containing 0.9–1.5 μg of purified RNA, using SuperScript IV and SuperScript First-strand synthesis kit (Invitrogen). 1–3 μl of reverse transcription products were used as templates in PCR reactions, except for *actin* RT-PCR which 1:10 diluted reverse transcription products were used. RT-PCR were performed with Taq DNA polymerase (NEB) at extension temperature of 68˚C. Gene/transcript specific primer pairs (5′−3′) and annealing temperatures are as follows: *plg* CGACATGGAGGTACACAAAG (Forward) and GAAGGACCTGCATGTAAAGG (Reverse), 49˚C; *pgk1*-FL AGGTCATTCTCATTCGGGAAC (F) and ACTGCAGCCTTGATTCTCTG (R), 51˚C; *pgk1*-alt AACTCATGCTAACACGGGGA (F) and ACTGCAGCCTTGATTCTCTG (R), 52.9˚C; *actin* AGGTCATCACCATCGGCAAT (F) and CAATGAAGGAAGGCTGGAACAG (R), 49 or 49.7˚C.

## In situ hybridization and immunohistochemistry

Whole mount in situ hybridization and immunohistochemistry protocols used in this study were previously described (*Phillips et al., 2001*). Whole mount stained embryos were cut into 10 μm sections using a cryostat as previously described (*Vemaraju et al., 2012*). The following antibodies were used in this study: Anti-Islet1/2 (Developmental Studies Hybridoma Bank 39.4D5, 1:100), Anti-GFP (Invitrogen A11122, 1:250), Alexa 546 goat anti-mouse or anti-rabbit IgG (ThermoFisher Scientific A-11003/A-11010, 1:50). Promega terminal deoxynucleotidyl transferase (M1871) was used according to manufacturer's protocol to perform the TUNEL assay.

## Cell transplantation

Wild-type donor embryos were injected with a lineage tracer (tetramethylrhodamine labeled, 10,000 MW, lysine-fixable dextran in 0.2 M KCl) at one-cell stage and transplanted into non-labeled wild-type or *pgk1* mutant embryos at the blastula stage. A total of 151 and 247 transplanted wild type cells were visualized in the ears of wild-type embryos and *pgk1* mutants, respectively, (n = 8 otic vesicles each) and analyzed for the expression of *etv5b*.

## Pharmacological treatments

The pharmacological inhibitors used in this study include Hexokinase inhibitor 2DG (2-Deoxy-Glucose, Sigma D8375), PFKFB3 inhibitor 3PO (Sigma SML1343), Pyruvate Dehydrogenase Kinase inhibitor DCA (dichloroacetate, Sigma 347795), Lactate Dehydrogenase inhibitor Galloflavin (Sigma SML0776), MCT1/2/4 inhibitors CHC (á-cyano-4-hydroxycinnamic acid, Sigma C2020) and UK-5099 (Sigma PZ0160), MEK inhibitor U1026 (Sigma U120) and PI3K inhibitor LY294002 (Sigma L9908). All inhibitors were dissolved in a 10 mm DMSO stock solution and diluted into the final indicated concentrations in fish water. Lactate treatments were performed with a 60% stock sodium DL-lactate solution (Sigma L1375) diluted 1 to 1000 in fish water to a final concentration of 6.7 mM, and buffered with 10 mm MES at pH 6.2. Note, lactate treatment has little effect on embryonic development at higher pH (not shown). Treatments were carried in a 24-well plate with a maximum of 15 embryos per well in a volume of 500 μl each.

## Statistics and quantitation

Student's t-test was used for pairwise comparisons. Analysis of 3 or more samples was performed by one-way ANOVA and Tukey post-hoc HSD test. Sufficiency of sample sizes was based on estimates of confidence limits. Sample sizes are indicated in Figures or legends. Hair cells were counted in whole mount specimens expressing *brn3c:GFP*, or by phalloidin-staining. Mature SAG neurons were counted by staining embryos with anti-Isl1/2 antibody and counting stained nuclei in whole mount preparations, or by counting *isl2b:GFP*+ SAG neurons in serial sections (2–4 embryos each) counter-stained with DAPI. For quantitation of gene expression patterns detected by in situ hybridization, stained cells were counted in serial sections (2–6 embryos each) counter-stained with DAPI. In some

cases, as indicated in the text, whole mount gene expression domains were quantified from by measuring cross-sectional areas using Photoshop. Otherwise characteristic changes in whole mount gene expression reported herein were fully penetrant amongst mutant embryos but were not observed in wild-type embryos.

## Whole genome sequencing and mapping analysis

Adult zebrafish males derived from the AB line were mutagenized using the alkylating agent N-ethyl-N-nitrosourea (ENU) and outcrossed to *Isl2b: GFP +* females as previously described (*Riley and Grunwald, 1995*). *sagd1* mutants were identified in the screening process due to the decreased expression of *Isl2b:Gfp* in vestibular SAG neurons. For the mapping analysis, *sagd1* mutants were outcrossed to the highly polymorphic zebrafish WIK line. A genomic pool of 50 homozygous mutants were collected from the intercrosses between 3 pairs of heterozygous *sagd1* carriers of the AB/WIK hybrid line and sent for whole genome sequencing at Genomic Sequencing and Analysis Facility (GSAF) at the University of Texas-Austin. Sequence reads were aligned using the BWA alignment software. Bulk segregant linkage and homozygosity mapping of the aligned sequences were performed using Megamapper (*Obholzer et al., 2012*). Both approaches identified the *pgk1* locus as the top candidate lesion in *sagd1* mutants.

## Acknowledgements

We thank Sarah Ferguson for her help in processing whole genome sequence data. This work was funded by NIH-NIDCD grant R01-DC03806.

## Additional information

### Funding

| Funder | Grant reference number | Author |
|---|---|---|
| National Institute on Deafness and Other Communication Disorders | R01-DC03806 | Bruce B Riley |

The funders had no role in study design, data collection and interpretation, or the decision to submit the work for publication.

### Author contributions

Husniye Kantarci, Data curation, Formal analysis, Validation, Investigation, Visualization, Methodology, Writing - original draft; Yunzi Gou, Data curation, Formal analysis, Investigation, Methodology, Writing - review and editing; Bruce B Riley, Conceptualization, Formal analysis, Supervision, Funding acquisition, Project administration, Writing - review and editing

### Author ORCIDs

Yunzi Gou (iD) http://orcid.org/0000-0002-3105-252X
Bruce B Riley (iD) https://orcid.org/0000-0001-6471-7965

### Ethics

Animal experimentation: This study was performed in strict accordance with the recommendations in the Guide for the Care and Use of Laboratory Animals of the National Institutes of Health. All of the animals were handled according to approved institutional animal care and use committee (IACUC) protocols (#2018-0124) of Texas A&M University.

### Decision letter and Author response

Decision letter https://doi.org/10.7554/eLife.56301.sa1
Author response https://doi.org/10.7554/eLife.56301.sa2

## Additional files

**Supplementary files**
• Source data 1. Raw data for cell counts.

• Transparent reporting form

### Data availability

All data generated or analyzed during this study are included in the manuscript and supporting files.

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
