## [Decision Letter]

**Acceptance summary:**

This paper uses genetic approaches in the zebrafish to tackle the under-appreciated interplay between metabolic and developmental signalling pathways in the developing embryo. The authors demonstrate a role for lactate synthesis in enhancing Fgf/MAPK signalling in the otic placode, precursor of the inner ear, and a requirement for these signalling pathways for correct sensory and neural development in this tissue.

**Decision letter after peer review:**

[Editors’ note: the authors submitted for reconsideration following the decision after peer review. What follows is the decision letter after the first round of review.]

Thank you for submitting your work entitled "The Warburg effect and lactate signaling augment Fgf signaling to promote sensory-neural development in the otic vesicle" for consideration by *eLife*. Your article has been reviewed by three peer reviewers, and the evaluation has been overseen by a Reviewing Editor and a Senior Editor. The following individuals involved in review of your submission have agreed to reveal their identity: Berta Alsina (Reviewer #1); David Raible (Reviewer #3).

Our decision has been reached after consultation between the reviewers. Based on these discussions and the individual reviews below, we regret to inform you that your work will not be considered further for publication in *eLife*.

As you can see, there was general consensus that while the initial observations are interesting and potentially important, there would be significant additional work necessary to meet the reviewers’ concerns. As we feel the additional work needed to address these issues would take more than two months to complete, we are returning your submission to you now in case you wish to submit elsewhere for speedy publication. However, if you address these points and wish to resubmit your work to *eLife*, we would be happy to look at a revised paper. Please note that it would be treated as a new submission with no guarantees of acceptance.

Reviewer #1:

The manuscript by Kantarci et al. deals with the activation of the lactate pathway during inner ear development. The authors uncover, through two different mutants, that loss of lactate synthesis reduces Fgf activity and affects hair cell formation and neurogenesis. The results are novel and interesting by linking for the first time a metabolic route with a classical developmental pathway. The authors by providing a wealth of different experiments and methodologies demonstrate the implication of this pathway in ear development.

1) The expression of *pkg1* is shown starting at 24hpf, where it is expressed in hair cells and mature SAG neurons. Since the strongest effects on *etv5*, *pax5* are observed at 18hpf and 24 hpf, it is not clear how this is correlated with pkg1 expression starting later than the effects and in restricted domains. It is very important that the early expression of pkg1 in the inner ear is shown since maybe the effects on early Fgf upregulation are not related directly to an ear phenotype but from other indirect effects.

2) The main rational of the paper for early effects of *sagd1* on Fgf activity and not later as well as an effect on the anterior (earlier development) but not posterior SAG, relies on the idea that *pgk1-alt* is required to upregulate *pgk1FL* and stabilize its mRNA. For this, the upregulation of *pgk1FL* should be shown better and in the inner ear. Moreover, the dynamics of mRNA decay in the mutant is not shown and could be shown by PCR. Mutagenic mutations in the STE of the pgk-alt could demonstrate that this is indeed the mode of action of *pgk1alt*.

3) On the other hand, the effects on hair cells and neurons is related to the expression of pkg1 in these cells? *Pgk1* does not seem to be expressed in the neurogenic domain but in mature neurons. How does loss of *Pgk1* affect neurogenesis at early time points?

Is *pgk1* expressed only in a subdomain of the SAG that could explain why only "vestibular" and not auditory neurons are affected? Please, show the expression of *pgk1* in the entire axis of the SAG.

I think that is misleading to talk about vestibular and auditory SAG, since the portion of the SAG that innervates the posterior macula, also innervates the posterior crista which is vestibular. Therefore, the separation into vestibular and auditory might be confusing in zebrafish and not as straight forward as in mammals because neurons innervating vestibular sensory organs are not segregated into different SAGs.

4) The authors indicate that other markers not related to Fgf signaling, such as Otx1b, dlx3b are not affected. However, these effects are only shown at 30hpf , when the authors mention that the effects of *pgk1* are minor. At this stage, *pax5* is also recovered, compared to the effect at 18hpf. Thus, to fully demonstrate that other markers in the inner ear are not changed and only Fgf signaling, the authors should show otx1b, dlx3b expression at 18hpf and Fgf ligands at these stages.

5) How are neurons counted in Figure 1? Are nuclei stained with DAPI to assess individual cells? With only cytoplasmatic staining of the isl2b reporter line, can individual cells be counted? Counting information is missing in MM. Which is exactly the posterior isl2b portion? The anterior SAG is outlined in Figure 1A but not posterior SAG. Are neurons counted though confocal sections? Again, it is not clear how counting is done as Figure 1C with overexposed images nor in Figure 2GN with ISH images of highly packed neurons.

6) If *pgk1* loss delays the upregulation of Fgf signaling, can the phenotype in anterior SAG or hair cells be recovered at later stages ( i.e 5 dpf?).

7) The authors discuss that the early effects are due to strong effect on Fgf upregulation early but not later, however, it is also possible that at later stages, other mechanisms compensate the lack of *pgk1alt* at later stages. Could it be that *pgk1alt* has only a maternal effect?

8) The loss of isl2b neurons in *pgk1* mutants (Figure 5H) is almost entire and affects anterior and posterior portions of the SAG. This phenotype seems stronger than the one of *sagd1* mutant. Is this so?

9) Can the authors show lactate secretion by using a fluorescent glucose substrate?

10) Authors state that Fgf ligand expression is not affected, but reduction in *Fgf3* is observable in Figure 5J. Could *pgk1* be upstream to Fgf signaling? On the other hand, does treatment of inner ear with SU5402 affect *pgk1* expression?

Reviewer #2:

The paper by Kantarci et al. focuses on the phenotype of mutant in which the *pgk1* gene is disrupted in zebrafish. The authors show that the mutant exhibits impaired Fgf-dependent development of hair cells and neurons in the otic vesicle and other neurons in the cranial ganglia and neural tube. The authors conclude that the crucial missing component in these mutant embryos is the loss of secreted lactate that is normally produced as a result of modified glycolysis that shunts pyruvate away from mitochondrial respiration in favor of lactate synthesis. They conclude that this secreted lactate activates the ERK MAPK pathway, which in turn primes the Fgf pathway to respond better to dynamic changes in Fgf.

The work is very interesting as it demonstrates that glycolytic enzymes that are usually thought of as housekeeping enzymes, actually play crucial roles in embryonic development.

In general the study is thorough, but there are several places where key experiments are missing and where more data are required.

1) Many of the experiments require quantitation. This has been done for some, but I think it should be done for all experiments where the authors want to conclude that gene expression is altered in a quantitative way. This is the case for Figure 3, Figure 4, and Figure 5. It is also crucial for the authors to indicate how many embryos they studied for each condition and how many showed the effects they are presenting. In addition, in Figure 2M and N. These data do not seem to match with the quantitations. Why is this?

2) Some of the in situs are rather poor quality and it is difficult to see the effect that the authors indicate. This is true of the in situs in Figure 3 and Figure 4S-X.

In Figure 3 it is not at all evident that sprouty 4 is affected by the *sagd1* mutation. Moreover, there is legend missing for Figure 3, and I think also labels missing in the Figure.

In Figure 4, it is not clear what is being shown in Figure 4S-Z. These images need to be much higher quality, and more clearly labeled to be able to draw any conclusions.

3) The authors conclude that the *sagd1* mutations are a result of disrupting the alternative spliced form of *pgk1* (*pgk1-alt*). This needs to be much more rigorously proven. What is the evidence in vivo that *pgk1-alt*, which has the exons 1a and 1b, also does not have the rest of FL *pgk1* downstream of exon 6? In other words, why do they think that only this transcript (ie exons1a, 1b 2-6) then splices to exon 6a. This needs to be proven. It is important that the authors show exactly what transcripts are normally present in the embryo.

They also should show that the *sagd1* mutations really do lead to no protein as a result of the mutated splice site or loss of the start codon.

Finally, they conclude that somehow the loss of *pgk1-alt* leads to destabilization of *pgk1-FL*. This needs to be demonstrated directly. It is much too speculative at the moment.

4) The model that the authors propose replies on the concept of lactate secretion. I think that a weakness of the paper is that this is assumed to occur, but not shown. This needs to be demonstrated directly. Also, the authors should demonstrate that this really does lead to upregulation of ERK MAPK, which is a key assumption in the paper.

Reviewer #3:

Kantarci et al. present an interesting study of changes in Fgf signaling and otic placode development after alterations in glycolysis. They describe the isolation and characterization of mutations affecting expression of *pgk1*. One allele, *sagd1*, affects an alternate transcript, suggesting a mechanism that regulates levels through a post-transcriptional mechanism. The authors present evidence that changes in glycolysis alter Fgf signaling.Together it is an interesting study that would potentially have broad appeal. There are several issues that need to be addressed:

The identification of the molecular nature of the *sagd1* mutant is not completely clear. The authors describe the nucleotide changes as background mutations in their stocks. The causality of these changes would be strengthened by linkage analysis genotyping individual embryos.

It is not clear what the evidence is for the *pgk1-alt* gene model. Has this been characterized by analysis of transcripts?

The authors should report whether the *hs:pgk1-alt* transgene is sufficient to rescue *sagd1* mutants. Following the author's logic, the *hs:pgk1-alt* transgene would not rescue *pgk1* mutants; this should also be tested.

Some clarity is needed for the phenotypes of *pgk1* mutants. Are they identical to sagd mutants – grossly normal, viable until 10-12d? Are there hair cell phenotypes in *pgk1* mutants? Are the trigeminal and reticulospinal phenotypes described for *pgk1* mutants also found in sagd mutants? Is there compensatory *pgk1* mRNA from maternal contributions?

Experiments testing the potential role of changes in plasmin in causing *pgk1* mutant phenotypes were incomplete. There were no positive controls for plasminogen morpholino oligonucleotides showing that plasmin levels were actually affected. There are sensitive commercial assays available to measure plasmin activity. Plasmin levels in wt and mutant animals should be measured, and levels after MO injection should be tested.

The authors hypothesize that Fgf and lactate converge on Mapk signaling. They should test this directly by assessing whether lactate treatment can partially rescue SAG neuron number after Mapk inhibitor treatment.

The authors should comment on whether the interference with lactate metabolism on Fgf signaling is specific, or whether other signaling pathways are also affected.

---

## [Author Response]

[Editors’ note: the authors resubmitted a revised version of the paper for consideration. What follows is the authors’ response to the first round of review.]

Reviewer #1:The manuscript by Kantarci et al. deals with the activation of the lactate pathway during inner ear development. The authors uncover, through two different mutants, that loss of lactate synthesis reduces Fgf activity and affects hair cell formation and neurogenesis. The results are novel and interesting by linking for the first time a metabolic route with a classical developmental pathway. The authors by providing a wealth of different experiments and methodologies demonstrate the implication of this pathway in ear development.1) The expression of pkg1 is shown starting at 24hpf, where it is expressed in hair cells and mature SAG neurons. Since the strongest effects on etv5, pax5 are observed at 18hpf and 24 hpf, it is not clear how this is correlated with pkg1 expression starting later than the effects and in restricted domains. It is very important that the early expression of pkg1 in the inner ear is shown since maybe the effects on early Fgf upregulation are not related directly to an ear phenotype but from other indirect effects.

We now show a time course of *pgk1* expression in Figure 3—figure supplement 5. We show that *pgk1* transcript is maternally supplied. Zygotic *pgk1* is broadly expressed at a low level but begins to upregulate in somites at 14 hpf, followed by upregulation in clusters of cells in the CNS and PNS beginning around 18 hpf and reaching maximum intensity by 24 hpf.

2) The main rational of the paper for early effects of sagd1 on Fgf activity and not later as well as an effect on the anterior (earlier development) but not posterior SAG, relies on the idea that pgk1-alt is required to upregulate pgk1FL and stabilize its mRNA. For this, the upregulation of pgk1FL should be shown better and in the inner ear. Moreover, the dynamics of mRNA decay in the mutant is not shown and could be shown by PCR.

Although *pgk1* expression does show regional upregulation, it is nevertheless expressed at a relatively low level (in the ear) that is not terribly clear in whole-mounts. We feel that sections of 24 hpf embryos shown in Figure 4 provide the clearest images for expression in the otic vesicle and SAG neurons. Figure 4 also shows that expression of *pgk1* is severely reduced in *sagd1* mutants at 24 hpf, when regional upregulation of *pgk1* is maximal. Earlier stages are less informative because the degree of upregulation in wild-type is still rather low. We also performed an RT-PCR analysis of *pgk1* transcript abundance in wt and mutants but observed no difference in amplicon intensity despite the clear reduction seen by wholemount in situ hybridization (Figure 3—figure supplement 4G and H). The likely reason for this is that upregulation in the CNS/PNS is highly localized and adds little to the overall basal expression in the rest of the embryo.

Mutagenic mutations in the STE of the pgk-alt could demonstrate that this is indeed the mode of action of pgk1alt.

Although we speculated in the Discussion that the STE of *pgk1-alt* might play a role in stabilizing *pgk1-FL* mRNA stability, we hope that reviewers will allow that testing this model via structure-function studies would entail a tremendous amount of work that we feel lies beyond the scope of this paper. We would need to show physical binding between *Pgk1-alt* protein and *pgk1-FL* transcript, perhaps in a pull-down experiment. Then we would need to test the effects of deleting the STE or other regions of *Pgk1-alt* on mRNA binding, as well as protein stability (since many mutations in human Pgk1 greatly accelerate its proteolytic turnover), and then validate results in vivo. Although this would be an interesting and illuminating exercise, we feel it would be worthy of an entire study by itself. However, in the spirit of the question we conducted new RT-PCR and knockdown studies (Figure 3—figure supplement 2, Figure 3—figure supplement 3, Figure 3—figure supplement 4 and Figure 4—figure supplement 4) that better validate *pgk1-alt* function.

3) On the other hand, the effects on hair cells and neurons is related to the expression of pkg1 in these cells? Pgk1 does not seem to be expressed in the neurogenic domain but in mature neurons. How does loss of Pgk1 affect neurogenesis at early time points?Is pgk1 expressed only in a subdomain of the SAG that could explain why only "vestibular" and not auditory neurons are affected? Please, show the expression of pgk1 in the entire axis of the SAG.

This question seems to suggest that *pgk1* acts cell-autonomously, but we performed a genetic mosaic experiment showing that *pgk1* acts non-autonomously (Figure 5K-R). Specifically, wild-type cells lying in the floor of the otic vesicle (within the neurogenic domain) in *pgk1-/-* mutant hosts do not properly express *etv5b* , despite their close proximity to sources of Fgf in the ear. Because *pgk1* and *Fgf* genes are coexpressed, and because blocking lactate transport phenocopies *pgk1-/-* mutants, the simplest interpretation is that Fgf and lactate are co-secreted, and both are required to act on nearby cells to fully activate the Fgf-Ras-MAPK pathway. Also as requested, Figure 4D-F’ shows that *pgk1* is expressed in both anterior and posterior regions of the SAG.

4) The authors indicate that other markers not related to Fgf signaling, such as Otx1b, dlx3b are not affected. However, these effects are only shown at 30hpf , when the authors mention that the effects of pgk1 are minor. At this stage, pax5 is also recovered, compared to the effect at 18hpf. Thus, to fully demonstrate that other markers in the inner ear are not changed and only Fgf signaling, the authors should show otx1b, dlx3b expression at 18hpf and Fgf ligands at these stages.

This is a good point, and we now show the earliest stages of otic expression of *fgf3* and *fgf8* in wild-type and *pgk1-/-* mutants (Figure 4—figure supplement 1). At 18 hpf, when *pax5* is first detected in the otic vesicle, *fgf3* is expressed in the hindbrain and pharyngeal endoderm but cannot be reliably detected in the ear. By 19 hpf both *fgf3* and *fgf8* expression are clearly detected in the ear and appear normal in *pgk1-/-* mutants, whereas *pax5* expression is strongly reduced in the mutant. As for regional markers, we now show that expression is normal in *pgk1-/-* mutants at 24 hpf (Figure 7). (Note, we focused on 24 hpf because several of these markers are not reliably expressed at earlier stages).

5) How are neurons counted in Figure 1? Are nuclei stained with DAPI to assess individual cells? With only cytoplasmatic staining of the isl2b reporter line, can individual cells be counted? Counting information is missing in MM. Which is exactly the posterior isl2b portion? The anterior SAG is outlined in Figure 1A but not posterior SAG. Are neurons counted though confocal sections? Again, it is not clear how counting is done as Figure 1C with overexposed images nor in Figure 2GN with ISH images of highly packed neurons.

The position of the posterior SAG is now marked by red outlines in Figure 1A. We have expanded Materials and methods to include an explanation for how counts were taken (see Statistics and Quantitation). For counting *isl2b:Gfp +* cells, or cells stained by ISH, cells were counted in serial sections counter-stained with DAPI. Anti-Isl1+ cells we counted from nuclear staining in wholemount embryos.

6) If pgk1 loss delays the upregulation of Fgf signaling, can the phenotype in anterior SAG or hair cells be recovered at later stages (i.e 5 dpf?).

We now present counts of hair cells and SAG neurons in wild-type embryos and *pgk1-/-* mutants at 5 dpf (Figure 5S, T). Deficiencies in both cell types are still evident at 5 dpf.

7) The authors discuss that the early effects are due to strong effect on Fgf upregulation early but not later, however, it is also possible that at later stages, other mechanisms compensate the lack of pgk1alt at later stages. Could it be that pgk1alt has only a maternal effect?

We provide new data showing that *pgk1-alt* transcript is not maternally supplied (Figure 3—figure supplement 4A). Additionally, we show that *pgk1-alt* splice-blocking MO (which only targets zygotic transcript) phenocopies *sagd1* and *pgk1-/-* mutants (Figure 3—figure supplement 3 and Figure 4—figure supplement 4B).

8) The loss of isl2b neurons in pgk1 mutants (Figure 5H) is almost entire and affects anterior and posterior portions of the SAG. This phenotype seems stronger than the one of sagd1 mutant. Is this so?

The reviewer is correct, both anterior and posterior portions of the SAG are deficient in *pgk1-/-* mutants, now shown in Figure 5S. The deficiencies are significant and persist through at least 5 dpf. Note that a slight deficiency of posterior SAG neurons is also seen in *sagd1* mutants, but differences were not statistically significant at any time point.

9) Can the authors show lactate secretion by using a fluorescent glucose substrate?

To our knowledge, fluorescent analogs of glucose (e.g. used to track glucose transport) cannot be metabolized into fluorescent analogs of lactate. Although a FRET sensor has been developed for tracking cytosolic lactate, none are currently available for tracking extracellular lactate. We note that there have been many studies suggesting that lactate secretion and uptake are widespread in the brain, but most of these studies relied on analysis of cells in tissue culture, or used indirect methods to infer changes in extracellular lactate in vivo. A few studies used micro-dialysis to directly measure extracellular lactate in adult brain tissue, but this approach cannot be applied to small local regions in zebrafish embryos. Thus we currently lack the ability to directly visualize or specifically measure extracellular lactate and are left only with indirect evidence (e.g. inhibitors of lactate transport phenocopy the mutants).

10) Authors state that Fgf ligand expression is not affected, but reduction in Fgf3 is observable in Figure 5J. Could pgk1 be upstream to Fgf signaling? On the other hand, does treatment of inner ear with SU5402 affect pgk1 expression?

The smaller domain of *fgf3* depicted in Figure 5J reflects that reduced size of sensory maculae in *pgk1-/-* mutants. New data presented in Figure 4—figure supplement 1 show that early expression of *fgf3* and *fgf8* appears normal in *pgk1-/-* mutants, and Figure 4—figure supplement 5 shows SU5402 does not alter expression of *pgk1*.

Reviewer #2:The paper by Kantarci et al. focuses on the phenotype of mutant in which the pgk1 gene is disrupted in zebrafish. The authors show that the mutant exhibits impaired Fgf-dependent development of hair cells and neurons in the otic vesicle and other neurons in the cranial ganglia and neural tube. The authors conclude that the crucial missing component in these mutant embryos is the loss of secreted lactate that is normally produced as a result of modified glycolysis that shunts pyruvate away from mitochondrial respiration in favor of lactate synthesis. They conclude that this secreted lactate activates the ERK MAPK pathway, which in turn primes the Fgf pathway to respond better to dynamic changes in Fgf.The work is very interesting as it demonstrates that glycolytic enzymes that are usually thought of as housekeeping enzymes, actually play crucial roles in embryonic development.In general the study is thorough, but there are several places where key experiments are missing and where more data are required.1) Many of the experiments require quantitation. This has been done for some, but I think it should be done for all experiments where the authors want to conclude that gene expression is altered in a quantitative way. This is the case for Figure 3, Figure 4, and Figure 5. It is also crucial for the authors to indicate how many embryos they studied for each condition and how many showed the effects they are presenting. In addition, in Figure 2M and N. These data do not seem to match with the quantitations. Why is this?

We now provide numbers of specimens studied for the images presented in each figure. Images in Figure 2M, N show *neurod* staining in the posterior SAG, which is not altered in *sagd1* mutants. It seems possible that the reviewer may have been looking at staining in other regions, so we added arrows to clarify which cells are in the SAG.

2) Some of the in situs are rather poor quality and it is difficult to see the effect that the authors indicate. This is true of the in situs in Figure 3 and Figure 4S-X.In Figure 3 it is not at all evident that sprouty 4 is affected by the sagd1 mutation. Moreover, there is legend missing for Figure 3, and I think also labels missing in the Figure.In Figure 4, it is not clear what is being shown in Figure 4S-Z. These images need to be much higher quality, and more clearly labeled to be able to draw any conclusions.

We acknowledge that some of the ISH images became washed-out when they were converted to pdf, making changes in expression level more difficult to discern. In re-building the pdf of the manuscript, inserted Figures were darkened to compensate, which makes changes easier to see. We apologize for accidentally truncating Figure 3 legend – this has been corrected. As for Figure 4S-Z, these data definitely confused the take-home message for Figure 4 and have been moved to other figures (Figure 3—figure supplement 4, Figure 3—figure supplement 5 and Figure 4—figure supplement 3). The new figures better document that additional Fgf-dependent cell types are deficient in pgk1-/- mutants (Figure 4—figure supplement 3) and show dynamic changes in transcript levels of *hs:pgk1-alt* vs. endogenous pgk1-FL following heat shock (Figure 3—figure supplement 5).

3) The authors conclude that the sagd1mutations are a result of disrupting the alternative spliced form of pgk1 (pgk1-alt). This needs to be much more rigorously proven. What is the evidence in vivo that pgk1-alt, which has the exons 1a and 1b, also does not have the rest of FL pgk1 downstream of exon 6? In other words, why do they think that only this transcript (ie exons1a, 1b 2-6) then splices to exon 6a. This needs to be proven. It is important that the authors show exactly what transcripts are normally present in the embryo.

The existence of *pgk1-alt* was confirmed by cloning and sequencing the cDNA obtained by RT-PCR amplification of mRNA expressed at 24 hpf. This is now explicitly stated in the Results, and some of the supporting data are shown in Figure 3—figure supplement 2. We do not claim that exons downstream of exon 6a are missing in *pgk1- alt* transcript, only that exon 6a contains an in-frame stop codon that presumably terminates translation of downstream sequence. Please note that Figure 4A shows that *pgk1-alt* retains exons 7-10, and this is now explicitly stated in the Results (subsection “Identification of the *sagd1* locus: A novel role for Pgk1”).

They also should show that the sagd1 mutations really do lead to no protein as a result of the mutated splice site or loss of the start codon.

We lack relevant antibodies that could identify *Pgk1-alt* peptide. However, we now show data on the effects of morpholinos that were designed to mimic the lesions detected in *sagd1* mutants (Figure 3—figure supplement 3). We show that a splice blocking MO (which blocks the splice acceptor in exon 1b) prevents accumulation of *pgk1-alt* transcript (Figure 3—figure supplement 3B) and phenocopies *sagd1* (Figure 3—figure supplement 3C). Two non-overlapping translation blockers were designed to bind either of the two AUG codons (tbMO1 and tbMO2) – these also phenocopy sagd1. Because tbMO2 binds too far downstream to block translation from the first AUG (containing the SNP found in *sagd1*), it is likely that the second AUG serves as a translation start site. Thus, the loss of splice acceptor in exon 1b is probably the relevant mutation in *sagd1*.

Finally, they conclude that somehow the loss of pgk1-alt leads to destabilization of pgk1-FL. This needs to be demonstrated directly. It is much too speculative at the moment.

We draw no conclusions about the mechanism of *pgk1-alt* function. Rather, we only conclude that *pgk1-alt* is necessary and sufficient for accumulation of *pgk1-alt* and *pgk1-FL* mRNAs. This implies that *pgk1-alt* regulates transcription or mRNA processing or stability. Although we do speculate in the Discussion that *pgk1-alt* could work by stabilizing *pgk1-FL* transcript, we feel this is a conservative suggestion drawn from published observations. Specifically, a number of previous studies (cited in the Discussion) show that mammalian Pgk1 protein binds specific mRNAs and can alter their stability; and translation of the STE sequence in *Pgk1* is required for its own mRNA stability. In contrast there are no published accounts indicating that Pgk1 regulates transcription. As mentioned above, we hope the reviewer will agree that conducting a detailed mechanistic here study would entail an enormous amount of work that would constitute an entire paper by itself and therefore lies beyond the scope of this paper.

4) The model that the authors propose replies on the concept of lactate secretion. I think that a weakness of the paper is that this is assumed to occur, but not shown. This needs to be demonstrated directly. Also, the authors should demonstrate that this really does lead to upregulation of ERK MAPK, which is a key assumption in the paper.

We agree that it would be ideal to directly demonstrate that lactate is secreted, but we know of no technology currently available that would enable us to demonstrate this. We do not assume that lactate must be secreted, nor does our summary model hinge on this issue. Instead, we suggest that lactate section is the simplest explanation for available data. Nevertheless, we added a statement in the Discussion acknowledging that we cannot exclude an alternative mechanism wherein lactate acts cell-autonomously to promote processing or secretion of Fgf ligands, although no such mechanisms have ever been reported (subsection “Enhanced glycolysis and lactate secretion promote Fgf signaling”).

We have attempted to use several commercial phospho-Erk antibodies to directly observe changes in MAPK activity, but we found all of them gave such high background staining that they were not useful. However, we have now shown that the Mekinhibitor U0126 blocks the stimulatory effect of exogenous lactate (Figure 6—figure supplement 2B). This confirms that MAPK is required for the stimulatory effects of lactate in our assays.

Reviewer #3:Kantarci et al. present an interesting study of changes in Fgf signaling and otic placode development after alterations in glycolysis. They describe the isolation and characterization of mutations affecting expression of pgk1. One allele, sagd1, affects an alternate transcript, suggesting a mechanism that regulates levels through a post-transcriptional mechanism. The authors present evidence that changes in glycolysis alter Fgf signaling. Together it is an interesting study that would potentially have broad appeal. There are several issues that need to be addressed:The identification of the molecular nature of the sagd1 mutant is not completely clear. The authors describe the nucleotide changes as background mutations in their stocks. The causality of these changes would be strengthened by linkage analysis genotyping individual embryos.

We included the original linkage analysis from homozygosity mapping that identified *pgk1* as our top candidate for *sagd1* (Figure 3—figure supplement 1). We also show in Figures 3—figure supplement 3 and Figure 4—figure supplement 4B that MOs designed to mimic the lesions detected in *sagd1* phenocopy the mutant.

It is not clear what the evidence is for the pgk1-alt gene model. Has this been characterized by analysis of transcripts?

We provide additional explanation of our initial cloning of *pgk1-alt* (using RT-PCR) and new RT-PCR experiments to clarify it’s function (Figures 3—figure supplement 3) and expression (Figure 3—figure supplement 2).

The authors should report whether the hs:pgk1-alt transgene is sufficient to rescue sagd1 mutants. Following the author's logic, the hs:pgk1-alt transgene would not rescue pgk1 mutants; this should also be tested.

We confirmed that *hs:pgk1-alt* cannot rescue *pgk1-/-* mutants (Figure 4—figure supplement 4A). Unfortunately, we were unable to test whether *hs:pgk1-alt* can rescue *sagd1* mutants because *sagd1* mutants have not been available during the last year. However, we performed an analogous experiment showing that *hs:pgk1-alt* rescues morphants injected with splice-blocking MO designed to mimic *sagd1* (Figure 4—figure supplement 4B).

Some clarity is needed for the phenotypes of pgk1 mutants. Are they identical to sagd mutants – grossly normal, viable until 10-12d? Are there hair cell phenotypes in pgk1 mutants? Are the trigeminal and reticulospinal phenotypes described for pgk1 mutants also found in sagd mutants? Is there compensatory pgk1 mRNA from maternal contributions?

We expanded the description of *pgk1-/-* mutants to better support the claim that they appear identical to *sagd1*. We now provide hair cell counts in Figure 5T. Figure 4 shows that *sagd1* mutants are deficient in trigeminal neurons, similar to the trigeminal deficiency seen in *pgk1-/-* mutants (now shown in Figure 4—figure supplement 3F). Figure 3—figure supplement 4 shows that *pgk1- FL* is maternally supplied, but not *pgk1-alt*. Regarding maternal function, we note that adding 2DG to *pgk1-/-* mutants does not exacerbate the phenotype, suggesting that residual maternal *pgk1* does not appreciably mitigate loss of zygotic function.

Experiments testing the potential role of changes in plasmin in causing pgk1 mutant phenotypes were incomplete. There were no positive controls for plasminogen morpholino oligonucleotides showing that plasmin levels were actually affected. There are sensitive commercial assays available to measure plasmin activity. Plasmin levels in wt and mutant animals should be measured, and levels after MO injection should be tested.

We performed new experiments with a splice-blocking MO, which reduces accumulation of *plg* mRNA but does not rescue the SAG deficiency in *pgk1-/-* mutants (Figure 4—figure supplement 6B, C).

The authors hypothesize that Fgf and lactate converge on Mapk signaling. They should test this directly by assessing whether lactate treatment can partially rescue SAG neuron number after Mapk inhibitor treatment.

This is a terrific idea. New data in Figure 6—figure supplement 2 show that the Mek inhibitor U0126 blocks the ability of lactate to stimulate SAG production. This is consistent with the idea that lactate acts though Raf kinase, upstream of Mek, and provides strong support for the hypothesis that lactate, like Fgf, feeds through the MAPK pathway.

The authors should comment on whether the interference with lactate metabolism on Fgf signaling is specific, or whether other signaling pathways are also affected.

We cannot exclude involvement of other pathways, but the ability of U0126 to mimic *pgk1-/-* mutants and to block the effects of lactate suggest that a deficiency of MAPK is sufficient to explain the phenotype.